# Distinct roles for CKM–Mediator in controlling Polycomb-dependent chromosomal interactions and priming genes for induction

Emilia Dimitrova ●[1] ✉, Angelika Feldmann ●[1,3], Robin H. van der Weide ●[2,4], Koen D. Flach[2], Anna Lastuvkova[1], Elzo de Wit ●[2] and Robert J. Klose ●[1] ✉

Precise control of gene expression underpins normal development. This relies on mechanisms that enable communication between gene promoters and other regulatory elements. In embryonic stem cells (ESCs), the cyclin-dependent kinase module Mediator complex (CKM–Mediator) has been reported to physically link gene regulatory elements to enable gene expression and also prime genes for induction during differentiation. Here, we show that CKM–Mediator contributes little to three-dimensional genome organization in ESCs, but it has a specific and essential role in controlling interactions between inactive gene regulatory elements bound by Polycomb repressive complexes (PRCs). These interactions are established by the canonical PRC1 (cPRC1) complex but rely on CKM–Mediator, which facilitates binding of cPRC1 to its target sites. Importantly, through separation-of-function experiments, we reveal that this collaboration between CKM–Mediator and cPRC1 in creating long-range interactions does not function to prime genes for induction during differentiation. Instead, we discover that priming relies on an interaction-independent mechanism whereby the CKM supports core Mediator engagement with gene promoters during differentiation to enable gene activation.

Mechanisms that shape 3D genome organization are thought to play important roles in controlling gene expression, particularly during development. For example, interactions between gene promoters, or gene promoters and other distal gene regulatory elements (like enhancers), have been implicated both in the maintenance of gene expression patterns and in enabling alterations in gene expression states during cell fate transitions[1–4].

A number of mechanisms have been proposed to create and regulate interactions between gene regulatory elements. For example, cohesin can extrude chromatin to establish topologically associated domains (TADs), which are generally constricted by CTCF-bound insulator sites. Cohesin-mediated loop extrusion is thought to increase the frequency of interactions between gene promoters and distal regulatory elements within TADs[5]. However, despite the profound effects that the disruption of cohesin or CTCF has on interactions within TADs, this typically translates into modest or tissue-specific effects on gene expression[6–10]. Although loop extrusion functions across the genome, other mechanisms are thought to play more direct and specific roles at gene regulatory elements by creating physical interactions that control gene expression. For example, the Mediator complex, which

[1]Department of Biochemistry, University of Oxford, Oxford, UK. [2]Division of Gene Regulation, Oncode Institute and The Netherlands Cancer Institute, Amsterdam, The Netherlands. [3]Present address: German Cancer Research Center (DKFZ), Heidelberg, Germany. [4]Present address: Hubrecht Institute KNAW, Utrecht, The Netherlands. ✉e-mail: emilia.dimitrova@bioch.ox.ac.uk; rob.klose@bioch.ox.ac.uk

is a fundamental regulator of gene transcription, has been proposed to support gene expression by functioning as a molecular bridge through binding transcription factors at active enhancers and RNA polymerase II at gene promoters[11–13]. However, recent work has questioned the extent to which the function of Mediator in gene expression relies on promoting physical interactions between regulatory elements[14–17]. At silent gene regulatory elements, binding of the Polycomb repressive complexes (PRCs) enables physical interactions between these inactive sites[18–28], which is thought to maintain gene repression[29–31] but may also poise genes for activation during cell lineage commitment[32–34]. In these contexts, whether chromosomal interactions themselves or other functions of the Polycomb system control gene expression is unknown. Thus, although it is evident that a variety of mechanisms have evolved to shape how gene regulatory elements physically interact with one another, the extent to which these interactions are required to control gene expression remains a central outstanding question[35–40].

The Mediator complex alone may not play a major role in enabling interactions between gene regulatory elements[14–17], but we and others have shown that a distinct form of the complex - which contains the cyclin-dependent kinase module CKM (composed of CDK8 or CDK19, CCNC, MED12 or MED12L, and MED13 or MED13L) and does not interact with RNA polymerase II[41–43] - is associated with gene regulatory element interactions in mouse ESCs[44–46]. Unlike the Mediator complex, CKM–Mediator (also known as CDK-Mediator) has been implicated in both repression and support of gene expression, suggesting that it might work through mechanisms that are distinct from the well-characterized function of Mediator in binding to and regulating RNA polymerase II activity[47,48]. In line with this possibility, CKM–Mediator appears to play specialized roles in controlling inducible gene expression after exposure to extracellular stimuli or cellular differentiation cues[47,49–54]. We and others have previously demonstrated that CKM–Mediator is recruited to the promoters of repressed developmental genes in ESCs[55–58] and this primes these genes for induction during differentiation[55]. In this context, CKM–Mediator binding appears to be important for creating interactions with other gene regulatory elements, suggesting that formation of 3D interactions may underpin its capacity to prime developmental genes for induction during cell lineage commitment[44].

Based on these findings, we set out to determine how CKM–Mediator controls chromosomal interactions and gene expression. To achieve this, we exploit inducible genetic perturbation systems and genomic approaches to examine CKM–Mediator function in ESCs and during cellular differentiation. We discover that CKM–Mediator contributes little to overall 3D genome organization in ESCs but is essential for creating interactions between Polycomb-bound regions of the genome. We show that CKM–Mediator does not define these interactions through an intrinsic bridging mechanism. Instead, it controls canonical PRC1 (cPRC1) binding at these sites, which in turn establishes contacts between Polycomb domains. Surprisingly, through separation-of-function experiments we reveal that Polycomb-dependent chromosomal interactions regulated by CKM–Mediator are not required for the priming or poising of genes for induction during differentiation. Instead, we discover that the priming function of CKM–Mediator relies on its ability to enable core Mediator binding to gene promoters during the process of gene induction.

## Results

### CKM–Mediator enables Polycomb domain interactions

To examine how CKM–Mediator influences genome organization in ESCs, we carried out in situ Hi-C in a cell line in which we can inducibly disrupt CKM–Mediator complex formation by removing its MED13/MED13L structural subunits (CKM–Mediator cKO; Fig. 1a,b and Extended Data Fig. 1a)[55]. No major alterations to overall genome organization were observed following CKM–Mediator disruption, with TADs and loop interactions remaining largely unchanged (Fig. 1c,d). It was previously proposed that CKM–Mediator promotes super enhancer-promoter interactions in ESCs[45,46]. However, we observed only subtle reductions in these interactions upon disruption of CKM–Mediator (Extended Data Fig. 1c). These data suggest that CKM–Mediator does not contribute centrally to 3D genome organization in ESCs.

ESCs are characterized by a unique set of extremely strong long-range interactions between regions of the genome that have high-level occupancy of PRCs, which we refer to as Polycomb domains[19–21,23,28,59]. These interactions are thought to contribute to developmental gene regulation either by maintaining repression in differentiated cell types or potentially by poising genes for induction during cell lineage commitment. Interestingly, a similar role in regulating developmental gene expression has been proposed for CKM–Mediator[44,55–57]. Given these seemingly similar functionalities, we asked whether CKM–Mediator might influence interactions between Polycomb domains. Remarkably, Hi-C analysis after CKM–Mediator disruption revealed dramatic reductions in interactions between Polycomb domains, and this effect was evident over a range of interaction distances (Fig. 1e,f and Extended Data Fig. 1d). A widespread reduction in Polycomb domain interactions was also observed using Capture-C analysis focused on promoters associated with Polycomb domains (Fig. 1g,h and Extended Data Fig. 1e,f). Therefore, CKM–Mediator is essential for interactions between Polycomb domains.

### CKM–Mediator supports cPRC1 binding to enable interactions

To understand how CKM–Mediator enables interactions between Polycomb domains, we asked whether CKM–Mediator is bound at these sites. The majority of Polycomb domains (91.12%) were enriched for the CKM–Mediator subunit CDK8, in general agreement with previous findings[56,57], suggesting that the effects of CKM–Mediator on Polycomb domain interactions may be direct (Fig. 2a and Extended Data Fig. 2a). It has previously been proposed that interactions between Polycomb domains are dependent on cPRC1, which in ESCs is defined by its structural subunit PCGF2 (refs. [20,24,30,60–65]). Given the profound effects on Polycomb domain interactions after the depletion of CKM–Mediator, we reasoned that CKM–Mediator may influence the function of cPRC1.

---

**Fig. 1 | CKM–Mediator has a limited role in 3D genome organization but is essential for Polycomb domain interactions. a**, A schematic of *Med13/13l*[fl/fl] ESCs. 4-Hydroxytamoxifen (TAM) induces conditional disruption of the CKM–Mediator complex (CKM–MED). **b**, A representative western blot analysis (*n* = 6) of nuclear extracts from *Med13/13l*[fl/fl] (wild type, WT) and *Med13/13l*[-/-] (CKM–MED KO) ESCs showing depletion of MED13 and MED13L proteins. HDAC1 is shown as a loading control. **c**, Hi-C contact matrices of WT and CKM–MED KO ESCs at 10 kb resolution. Genomic coordinates are indicated. **d**, Aggregate analysis of TADs and loops in WT and CKM–MED KO ESCs at 10 kb resolution. **e**, Hi-C contact matrices of WT and CKM–MED KO ESCs at 5 kb resolution. Interactions between Polycomb domains are indicated with a red circle. The blue track shows binding of PRC1 (RING1B ChIP–seq). Genomic coordinates are indicated. **f**, Aggregate analysis of Hi-C signal (10 kb resolution) at pairs of Polycomb domains in *Med13/13l*[fl/fl] (WT) and *Med13/13l*[-/-] (CKM–MED KO) ESCs, with 200 kb flanking regions. The difference between WT and KO is shown. **g**, A snapshot showing Capture-C read count signal in WT and CKM–MED KO ESCs. Interactions between the *Nkx2-1* promoter bait (triangle) and surrounding Polycomb-bound sites are shown with arrowheads. PRC1 binding (RING1B ChIP–seq) is shown as a reference. **h**, Boxplot analysis of mean normalized read counts from WT and CKM–MED KO ESCs showing interactions between Polycomb gene promoters and other Polycomb domains (left), or non-Polycomb gene promoters and active sites (H3K27ac, right). Interactions were not distance-matched due to differences in the interaction ranges for the two promoter types. Boxes show IQRs, center line represents median, whiskers extend by 1.5 × IQR or the most extreme point (whichever is closer to the median), while notches extend by 1.58 x IQR/sqrt(*n*), giving a roughly 95% confidence interval for comparing medians.

To test this possibility, we examined cPRC1 occupancy after CKM–Mediator disruption by carrying out calibrated ChIP–seq (cChIP–seq) using antibodies recognising the cPRC1 subunits RING1B, PCGF2 and CBX7. Importantly, this revealed a major reduction in cPRC1 binding at

Polycomb target sites in the absence of CKM–Mediator (Fig. 2b-d and Extended Data Fig. 2b,c), despite only subtle reductions in protein levels (Extended Data Fig. 2d). cPRC1 associates with Polycomb domains via its CBX7 subunit that binds H3K27me3 deposited by PRC2 (refs. [66–68]).

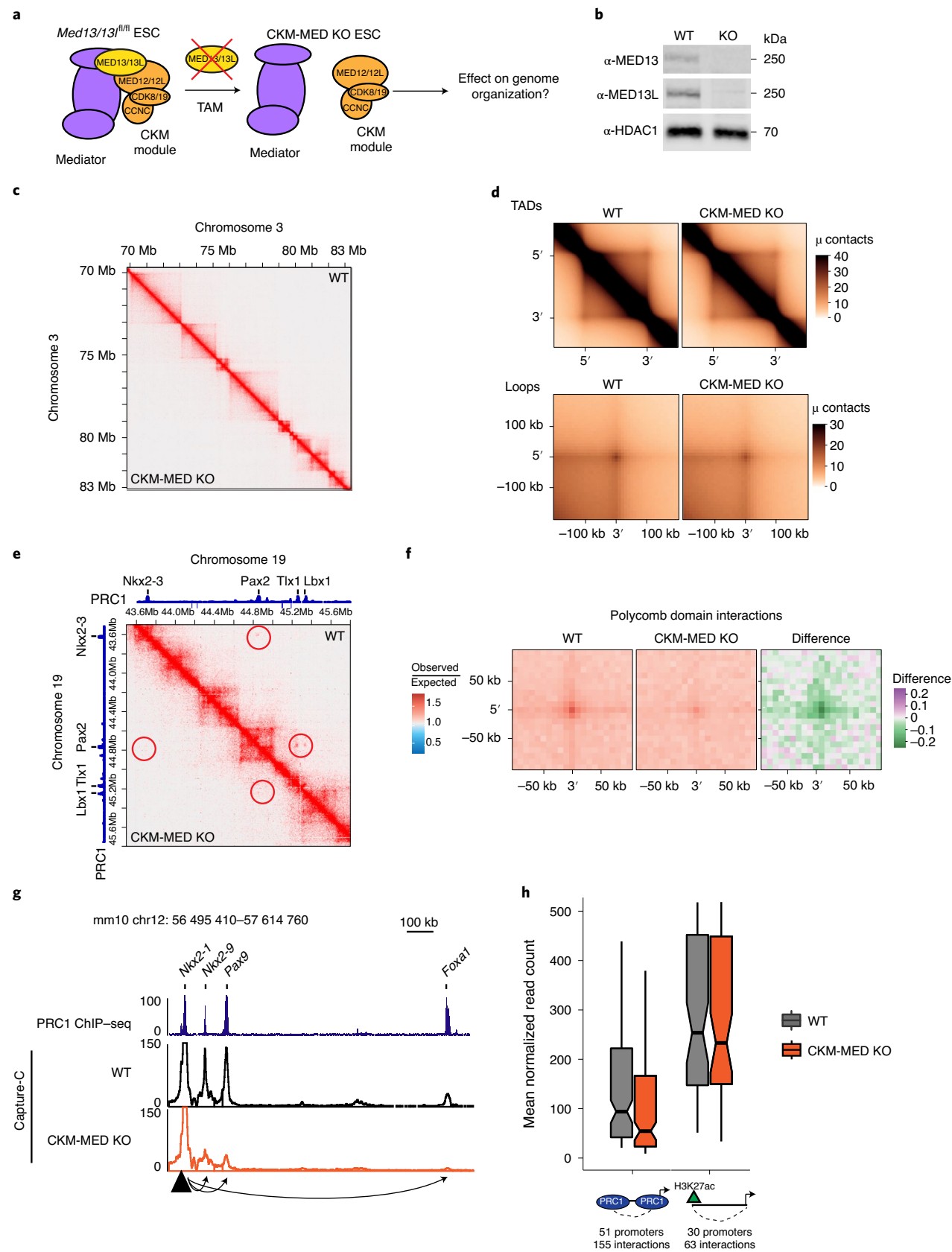

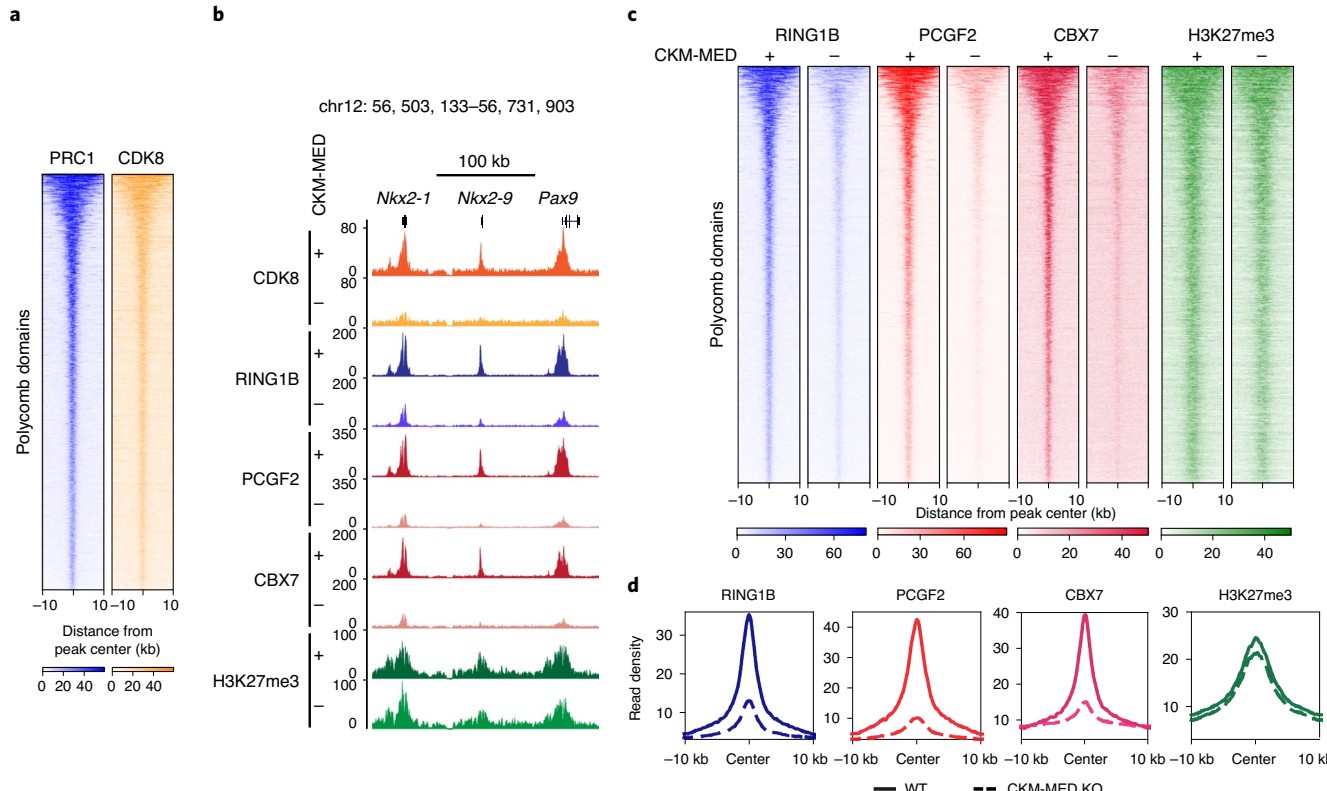

**Fig. 2 | CKM–Mediator regulates canonical PRC1 binding. a**, Heatmaps showing RING1B (PRC1) and CDK8 ChIP–seq signals at Polycomb domains (*n* = 2097), sorted by decreasing RING1B signal. **b**, A genomic snapshot of a Polycomb-bound locus, showing CDK8, RING1B, PCGF2, CBX7 and H3K27me3 ChIP–seq signal in WT (+) and CKM–MED KO (-) ESCs. **c**, Heatmaps showing RING1B, PCGF2, CBX7 and H3K27me3 ChIP–seq signal at Polycomb domains (*n* = 2,097) in WT (+) and CKM–MED KO (-) ESCs, sorted by decreasing RING1B signal. **d**, Metaplot analysis of RING1B, PCGF2, CBX7 and H3K27me3 enrichment at Polycomb domains (*n* = 2,097) in WT and CKM–MED KO ESCs.

Interestingly, cChIP–seq for H3K27me3 revealed only modest reductions in this modification after CKM–Mediator disruption (Fig. 2b-d). Therefore, CKM–Mediator regulates cPRC1 binding without major effects on H3K27me3.

Given that cPRC1 has been proposed to enable interactions between Polycomb domains[20,24,30], and its binding is abrogated following disruption of CKM–Mediator (Fig. 2), the observed effect on Polycomb domain interactions in the absence of CKM–Mediator may be due to loss of cPRC1 occupancy. In agreement with this possibility, the effects on cPRC1 binding were related to the reductions in interactions after depletion of CKM–Mediator (Extended Data Fig. 2e) and corresponded to the level of CKM–Mediator binding (Extended Data Fig. 2f). However, CKM–Mediator has also been proposed to function as a molecular bridge to enable chromosomal interactions[44–46]. Given that both cPRC1 and CKM–Mediator binding are lost upon CKM–Mediator disruption, interactions could be defined by either cPRC1 or CKM–Mediator. To discover the molecular determinant that enables these interactions, we took advantage of a synthetic system to create a separation-of-function scenario in which either cPRC1 or CKM–Mediator could be ectopically tethered to an artificial site in the genome[69] (Fig. 3a and Extended Data Fig. 3a,b). Importantly, tethering CDK8 recruited CKM–Mediator and tethering PCGF2 recruited the cPRC1 complex[69] but not CKM–Mediator (Extended Data Fig. 3c). We then asked whether binding of cPRC1 or CKM–Mediator at this ectopic site was able to support interactions with nearby regions co-occupied by cPRC1 and CKM–Mediator. These data revealed that cPRC1 was sufficient to create de novo interactions with surrounding sites, in line with similar findings from PRC2 tethering[70], which would lead to recruitment of cPRC1 (ref. [69]). By contrast, we found no evidence for

interactions with surrounding sites when CKM–Mediator was tethered (Fig. 3b and Extended Data Fig. 3d). Importantly, endogenous control sites retained interactions in both cell lines, although they were slightly weaker in the CKM–Mediator tethered line (Extended Data Fig. 3e).

To further explore whether cPRC1 is the central determinant underpinning Polycomb domain interactions, we next used a cell line in which we can inducibly disrupt the cPRC1 complex by removing the core structural components PCGF2 and PCGF4 (cPRC1 cKO)[59] and carried out Capture-C (Fig. 3c and Extended Data Fig. 3f). Importantly, removal of cPRC1 caused a near complete loss of interactions between Polycomb domains, while most sites retained CKM–Mediator binding (Fig. 3d,e and Extended Data Fig. 3g-i). Therefore, cPRC1 establishes long-range interactions between Polycomb domains, with CKM–Mediator playing a regulatory role in facilitating cPRC1 binding.

## cPRC1 interactions are not required for CKM–Mediator priming

CKM–Mediator occupies silent developmental gene promoters in ESCs and is required for subsequent gene activation during differentiation[55,56]. In some cases CKM–Mediator occupancy corresponds to pre-formed long-range interactions with other gene regulatory elements, suggesting that by bringing gene regulatory elements in close proximity with each other in ESCs, CKM–Mediator may prime them for future activation[44]. We now show that CKM–Mediator-dependent interactions are reliant on cPRC1 (Fig. 3). Importantly, the Polycomb system has similarly been implicated in poising or priming genes for activation during differentiation by creating interactions between gene promoters and other regulatory elements, including poised enhancers[32,33]. Based on this functional convergence between CKM–Mediator

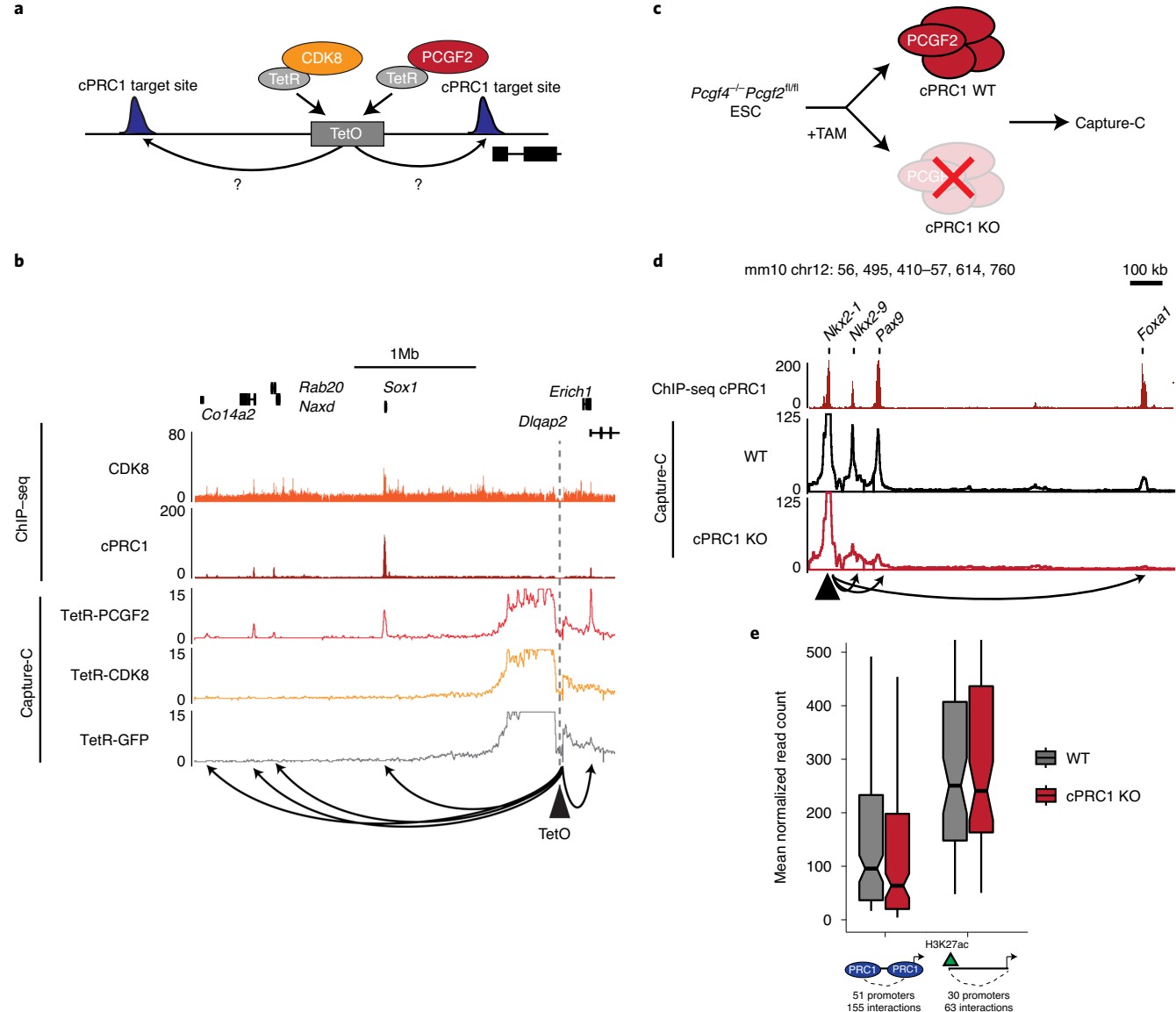

**Fig. 3 | cPRC1 creates interactions between Polycomb domains. a**, A schematic of the integrated TetO site and experimental setup. **b**, A snapshot showing Capture-C read count signal from TetR-PCGF2, TetR-CDK8 and TetR-GFP lines at the TetO array. CDK8 and PCGF2 (cPRC1) ChIP–seq signal is given as a reference. The TetO bait is shown as a triangle and interactions created with surrounding cPRC1-bound sites are represented with arrowheads. **c**, A schematic of the cPRC1 (*Pcgf4⁻/⁻ Pcgf2*ᶠˡ/ᶠˡ) conditional KO line. **d**, A snapshot showing Capture-C read count signal from WT and cPRC1 KO ESCs. Interactions between the *Nkx2-1* promoter bait (triangle) and surrounding Polycomb domain sites are shown

with arrowheads. cPRC1 binding (PCGF2 ChIP–seq) is shown as a reference. **e**, Boxplot analysis of normalized read counts from WT and cPRC1 KO ESCs showing interactions between Polycomb gene promoters and other Polycomb domains (left), or non-Polycomb gene promoters and active sites (H3K27ac, right). Boxes show IQR, center lines represent the median, whiskers extend 1.5 × IQR or the most extreme point (whichever is closer to the median), whereas notches extend by 1.58 x IQR/sqrt(*n*), giving a roughly 95% confidence interval for comparing medians.

and cPRC1 activities, we hypothesized that CKM–Mediator may enable interactions between gene promoters and regulatory elements via a cPRC1-dependent mechanism to prime genes for activation during differentiation.

To examine this possibility, we used all-*trans* retinoic acid to drive ESC differentiation and carried out calibrated nuclear RNA-seq (cnRNA-seq) to identify genes that rely on CKM–Mediator for their induction during differentiation (Fig. 4a and Extended Data Fig. 4a,b). Based on this analysis, 631 CKM–Mediator-dependent genes (fold change of > 1.5, adjusted *P* value < 0.05) were identified (Fig. 4b and Extended Data Fig. 4c). Importantly, these genes also showed cPRC1 enrichment at their promoters in ESCs and PRC1 occupancy tended

to be reduced following RA treatment (Extended Data Fig. 4d,e). To determine whether cPRC1 and its capacity to mediate chromosomal interactions enables gene induction by CKM-Mediator, we depleted cPRC1 and induced differentiation (Fig. 4c and Extended Data Fig. 4f,g). On average, CKM–Mediator-dependent genes induced normally in the absence of cPRC1 (Fig. 4d,e and Extended Data Fig. 4h), with only 18 of these genes showing a significant decrease in activation (Extended Data Fig. 4h,i). Therefore, while CKM–Mediator contributes to gene induction, it does not seem to do so through a cPRC1-dependent mechanism.

This finding prompted us to investigate more generally whether cPRC1 has a role in gene induction during differentiation, particularly of genes that engage in interactions. Therefore, our analysis was extended

to include RA-induced genes that are part of a previously described Polycomb interaction network in ESCs (*n* = 482) (Extended Data Fig. 4j,k)[19]. Interactions between RA-induced genes were lost in the absence of cPRC1, including interactions with poised enhancers (Extended Data Fig. 4l,m). However, as with CKM–Mediator-dependent genes, this had minimal effect on gene induction (Fig. 4e,f and Extended Data Fig. 4n). In contrast, we identified 184 genes within the Polycomb interaction network that rely on CKM–Mediator for induction (Fig. 4f). Therefore, CKM–Mediator has an essential role in gene activation during differentiation, independent of cPRC1-mediated chromosomal interactions. Furthermore, cPRC1 does not poise genes for activation during differentiation, despite its role in enabling interactions between gene promoters and other regulatory elements in ESCs.

### CKM–Mediator primes genes by enabling core Mediator binding

CKM–Mediator is essential for enabling cPRC1 to create interactions between Polycomb domain-associated gene regulatory elements, but these interactions are dispensable for gene induction during differentiation. In the absence of a pre-formed interaction mechanism for priming, we hypothesized that CKM–Mediator may prime genes for activation during differentiation by more directly influencing the function of the core Mediator[71–73]. To investigate this possibility, we engineered an epitope tag into the endogenous *Med14* gene, which is a structural subunit of the core Mediator (Extended Data Fig. 5a-c). Addition of the epitope tag did not interfere with CKM–Mediator complex formation (Extended Data Fig. 5d) and, therefore, enabled us to carry out ChIP–seq analysis and examine core Mediator occupancy in ESCs and during differentiation. Despite high levels of CDK8 binding at the promoters of CKM–Mediator-dependent genes (Fig. 5) and, more broadly, over Polycomb domains in ESCs (Extended Data Fig. 5e), the occupancy of MED14 at these sites was much lower than at active sites (Extended Data Fig. 5e). This suggests that, although the CKM–Mediator can bind to inactive developmental gene promoters, binding of the core Mediator may be more dynamic at these sites. Furthermore, it raised the interesting possibility that the mechanism of core Mediator binding and its stability at activated sites could change during the process of gene activation, so that it enters into a state that relies less on the CKM for engagement, as has been suggested previously[58].

Based on these observations, we were keen to examine core Mediator association with these sites during differentiation. During the transition to an active state, promoters of CKM–Mediator-dependent genes showed reduced levels of CDK8 binding, and they accumulated more MED14 (Fig. 5 and Extended Data Fig. 5f,g). We then asked whether the CKM module was required for this increased association of the core Mediator during differentiation[12,13,58]. Indeed following RA induction, promoters of CKM–Mediator-dependent genes do not acquire more MED14 in the absence of the CKM module (Fig. 5a,b and Extended Data Fig. 5g), consistent with these genes failing to induce appropriately (Fig. 4). Therefore, we propose that the CKM module primes genes for induction, not by pre-forming 3D gene regulatory interactions through the Polycomb system, but instead by enabling efficient engagement of the core Mediator at target gene promoters to support transcription activation during differentiation.

## Discussion

To define the extent to which interactions between gene regulatory elements are required for controlling gene expression has been challenging. This is due to the fact that many of the proteins and complexes that are proposed to enable these interactions are also known to have direct roles in controlling transcription at gene promoters. Here, we show that CKM–Mediator contributes very little to 3D genome organization in ESCs but is specifically required for interactions between Polycomb-bound gene regulatory elements (Fig. 1). These interactions do not rely directly on a CKM–Mediator-based bridging mechanism (Fig. 3), but instead CKM–Mediator controls binding of the cPRC1 complex (Fig. 2) to enable interactions between Polycomb domains (Fig. 3). By removing cPRC1, we specifically disrupt these interactions and reveal that CKM–Mediator is still able to prime genes for activation during differentiation (Fig. 4) through supporting recruitment of the core Mediator to gene promoters (Fig. 5). Therefore, CKM–Mediator primes genes for activation during differentiation by supporting recruitment of the core Mediator.

Physical interactions between gene regulatory elements are thought to enable gene expression[32,44,74,75]. In line with this concept, it has been proposed that, through the function of Polycomb and/or CKM–Mediator complexes, pre-formed interactions that tether silent developmental genes and other regulatory elements in stem cells may render genes poised or primed for activation during differentiation[32–34,44,76,77]. Here, we demonstrate that pre-formed interactions between gene regulatory elements co-occupied by CKM–Mediator and cPRC1 rely on cPRC1, and that the binding of cPRC1 is regulated by CKM–Mediator. Although the precise mechanisms through which CKM–Mediator facilitates cPRC1 binding to create interactions remain an open question for further study this realization allowed us to create a separation-of-function scenario whereby we could disrupt pre-formed interactions by removing cPRC1 yet leave CKM–Mediator intact. Importantly, in the context of these experiments, we find no evidence to suggest that pre-formed regulatory interactions play a prominent role in priming genes for activation during differentiation. Consistent with these findings, studies have shown that cPRC1 does not contribute to gene regulation during embryoid body formation in vitro[18], and cPRC1-null mice develop normally until 8.5 dpc, by which point a host of key developmental gene expression transitions have already been completed[78,79].

Instead, we find that the CKM–Mediator appears to have a more direct role in priming genes for induction during differentiation by ensuring appropriate association of the core Mediator complex during activation. This priming is likely to involve FBXL19, which physically interacts with CKM–Mediator and recruits CKM–Mediator to silent developmental gene promoters by binding to CpG-island DNA[55]. We speculate that pre-binding of CKM–Mediator might provide transcriptional activators with a localized pool of core Mediator that can be co-opted to support the timely induction of silent developmental genes during cellular differentiation. However, other related models could be envisaged that explain the mechanics of priming, including transcriptional activators evicting the CKM from pre-bound CKM–Mediator to enable transition of silent developmental genes into an activated state. Given the dynamic nature of these systems in vivo[80,81], it is extremely difficult to distinguish between these related yet distinct

**Fig. 4 | CKM–Mediator primes genes for activation during differentiation independently of cPRC1-mediated interactions. a**, A schematic of the differentiation of WT and CKM–MED KO ESCs used for cnRNA-seq. **b**, Boxplot analysis of the expression of CKM–MED-dependent genes (*n* = 631) in WT ESCs and following RA (retinoic acid) induction (WT and CKM–MED KO). Boxes show IQR, center lines represent the median, whiskers extend by 1.5 × IQR or the most extreme point (whichever is closer to the median), whereas notches extend by 1.58 x IQR/sqrt(*n*), giving a roughly 95% confidence interval for comparing medians. **c**, A schematic of the differentiation of WT and cPRC1 KO ESCs for

cnRNA-seq. **d**, As in **b** but for cPRC1 cKO cells. **e**, A screenshot showing the expression of genes within the *HoxB* cluster following RA induction of CKM–MED cKO or cPRC1 KO cells. Forward strand is shown on top and reverse strand is shown at the bottom of each track. ChIP–seq tracks for CDK8 and cPRC1 (PCGF2) enrichment are shown. **f**, Boxplot analysis of the expression of RA-induced (RA-ind) genes from the Polycomb (PcG) network (top, n=482) and CKM–Med-dependent genes from the PcG network (bottom, n=184) following RA induction of CKM–MED cKO or cPRC1 KO cells. Boxes are defined as in **a**.

biochemical models. In future work, kinetic experiments using rapid degron approaches may help to resolve these points and also provide insight into how CKM–Mediator influences cPRC1 binding. However,

consistent with the requirement for pre-binding of CKM–Mediator in priming genes for induction, removal of FBXL19 causes a reduction in CKM binding at silent developmental gene promoters and, similarly to

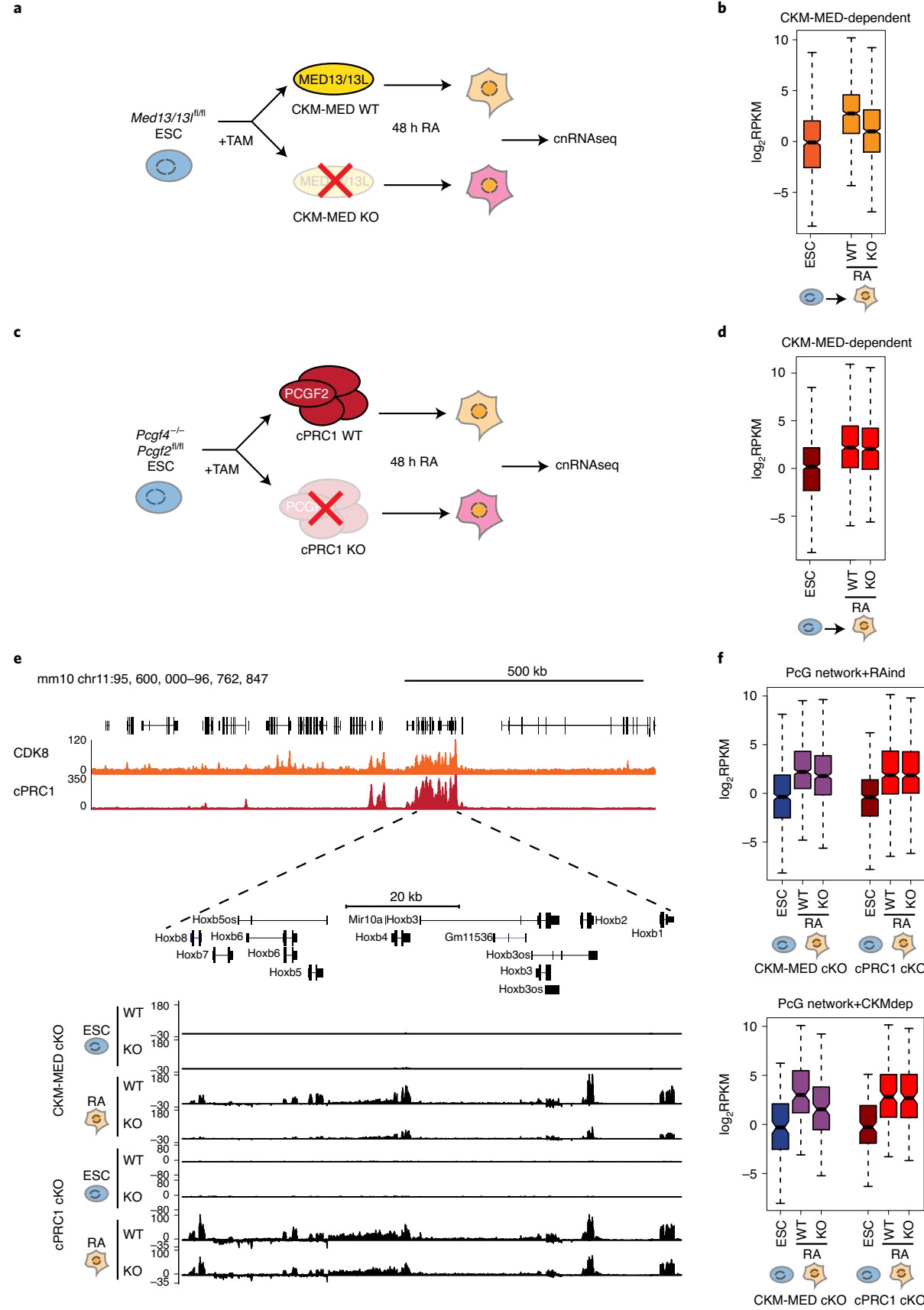

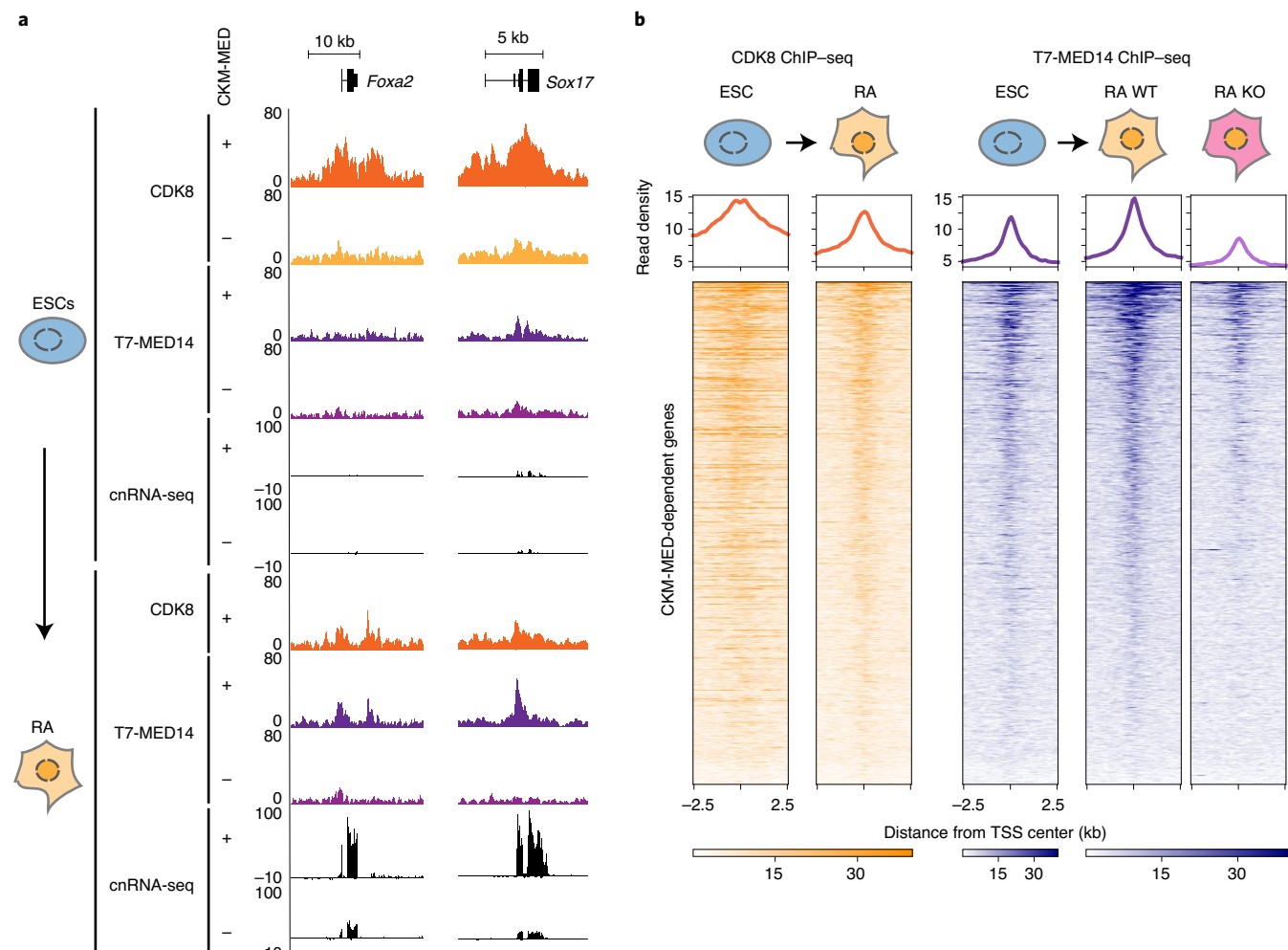

**Fig. 5 | CKM–Mediator enables gene induction via recruitment of the Mediator complex. a**, A genomic snapshot of two CKM-Mediator-dependent genes, showing CDK8 and T7-MED14 ChIP–seq and cnRNA-seq in WT (+) and CKM–MED KO (-) ESCs (top) and following RA induction (bottom). **b**, Heatmaps showing CDK8 and T7-MED14 ChIP–seq signal at promoters (TSS±2.5 kb) of CKM–MED-dependent genes in ESCs and following RA induction (*n* = 631). T7-MED14 signals are shown for WT and CKM–Mediator KO RA-induced cells. Genes are sorted by decreasing T7-MED14 signal in RA-treated cells. Metaplots showing read density are shown on the top of each heatmap.

CKM–Mediator removal, renders them less competent for induction during differentiation[55]. Furthermore, mice deficient for CKM subunits display pre-implantation lethality, consistent with an essential role in early developmental gene expression transitions[82–84]. As such, CKM–Mediator appears to function to prime genes for induction through supporting core Mediator acquisition at gene promoters during gene induction, not through mechanisms that create pre-formed regulatory element interactions.

These new findings raise the important question of why CKM–Mediator regulates cPRC1 binding to create interactions between silent gene regulatory elements if this is not related to its role in priming genes for induction during differentiation. A hint as to why this might be important comes from genetic screens in Drosophila, in which the CKM–Mediator complex components MED12 and MED13 were identified as Polycomb group genes that enable the long-term maintenance of Hox gene repression[85]. In agreement with a potential repressive role for CKM–Mediator at Polycomb target genes, it was recently shown that the CDK8 component of the CKM–Mediator complex has important roles in maintaining X-chromosome inactivation in mice[86] and that CDK8 absence leads to loss of Polycomb-mediated gene silencing[86,87]. Interestingly, in both of these scenarios, CKM–Mediator and Polycomb appear to maintain repression in more differentiated cells, whereas, in contrast, cPRC1 disruption has little effect on the maintenance of Polycomb target gene repression in ESCs[18,59,88]. As such, we envisage that the role that CKM–Mediator plays in regulating cPRC1 occupancy to create long-range interactions between silent regulatory elements may be particularly important in maintaining long-term gene repression in more differentiated cell types, yet contribute less to gene repression in rapidly dividing stem cells. This is consistent with the observation that cPRC1-deficient mice display inappropriate maintenance of Polycomb target gene repression and lethality in later embryonic stages[78,79].

Based on its seemingly distinct roles in gene regulation, we propose that CKM–Mediator may play a 'yin-and-yang' role in controlling expression. We hypothesize that during early developmental stages CKM–Mediator associates with silent developmental gene promoters to support gene induction during differentiation by helping to enable core Mediator binding during the transition to an activated state. However, in the absence of an activation signal at later developmental stages, the distinct role of CKM–Mediator in enabling cPRC1 binding to create interactions with other silent Polycomb-occupied regulatory sites could predominate to help maintain long-term gene repression. As such, distinct CKM–Mediator functions could play

important and complementary roles in supporting developmental gene regulation. In future work, it will be important to test these new models for CKM–Mediator function in appropriate mouse developmental model systems.

In summary, we show that CKM–Mediator is essential for regulating interactions between Polycomb domains. However, these interactions contribute little to gene activation during differentiation. Instead, we show that CKM–Mediator primes genes for induction during differentiation by supporting core Mediator binding to promoters during gene activation.

## Online content

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

## Methods

### Cell culture

Mouse ESCs were cultured on gelatin-coated dishes (Sigma-Aldrich) in DMEM (Thermo Fisher Scientific) supplemented with 15% fetal bovine serum (BioSera), 2mM L-glutamine, 0.5 mM beta-mercaptoethanol, 1× non-essential amino acids, 1× penicillin-streptomycin solution (Thermo Fisher Scientific) and 10 ng ml⁻¹ leukemia-inhibitory factor (produced in-house). *Med13/13l*ᶠˡ/ᶠˡ ERT2-Cre[55] and *Pcgf4*⁻/⁻/*Pcgf2*ᶠˡ/ᶠˡ ERT2-Cre ESCs[59] were treated with 800 nM 4-hydroxytamoxifen (Sigma-Aldrich) for 96 h and 72 h, respectively. For RA differentiation of ESCs, $4 \times 10^6$ ESCs were allowed to attach to gelatinized 15 cm dishes for 6–8 h and treated with 1 μM all-*trans* retinoic acid (Sigma-Aldrich) in EC-10 medium (DMEM supplemented with 10% fetal bovine serum, L-glutamine, beta-mercaptoethanol, non-essential amino acids and penicillin-streptomycin) for 48 h. TOT2N E14 ESCs used for TetR targeting experiments were previously described[69]. To generate TetR-CDK8 TOT2N ES lines, TOT2N E14 ESCs were transfected using Lipofectamine 2000 (Thermo Fisher Scientific) following the manufacturer's instructions. Stably transfected cells were selected for 10 days using 1 μg ml⁻¹ puromycin, and individual clones were isolated and expanded in the presence of 1 μg ml⁻¹ puromycin to maintain transgene expression. HEK293T cells, used for calibration of crosslinked cChIP−seq experiments, were cultured in EC-10 media. All mammalian cell lines were cultured at 37 °C and 5% CO₂. SG4 *Drosophila* cells, used for calibration of ncRNA-seq and native ChIP−seq experiments, were grown at 25 °C in Schneider's medium (Thermo Fisher Scientific) supplemented with 10% heat-inactivated fetal bovine serum (BioSera) and penicillin-streptomycin. All cell lines generated and grown in the Klose Lab were routinely tested for mycoplasma infection.

### Generation of the MED14-T7 *Med13/13l*ᶠˡ/ᶠˡ ESC line

To allow for efficient chromatin immunoprecipitation of MED14, we introduced an amino-terminal 3xT7-2xStrepII-FKBP12 tag to the endogenous *Med14* gene. The tag was synthesized by GeneArt (Thermo Fisher Scientific). The targeting construct was generated by Gibson assembly (Gibson Assembly Master Mix kit, New England Biolabs) of the PCR-amplified tag sequence and roughly 520 bp homology arms surrounding the ATG start codon of the *Med14* gene, amplified from mouse genomic DNA.

The pSptCas9(BB)-2A-Puro(PX459)-V2.0 vector was obtained from Addgene (no. 2988) and the sgRNA was designed using the CRISPOR online tool (http://crispor.tefor.net/crispor.py). The targeting construct was designed such that the endogenous ATG sequence is absent, and the Cas9 recognition site is disrupted by the insertion of the tag. ESCs were transfected in a single well of a 6-well plate with 0.5 μg Cas9 guide plasmid and 2 μg targeting construct plasmid using Lipofectamine 3000 (Thermo Fisher Scientific) according to the manufacturer's guidelines. The day after transfection, cells were passaged at a range of densities and subjected to puromycin selection (1 μg ml⁻¹) for 48 h. Approximately 7–10 days following transfection, individual clones were isolated, expanded and PCR-screened for the homozygous presence of the tag.

### Preparation of nuclear extracts and Western blot analysis

Collected cells were resuspended in 10 × pellet volume (PV) of Buffer A (10 mM Hepes pH 7.9, 1.5 mM MgCl2, 10 mM KCl, 0.5 mM DTT, 0.5 mM PMSF, cOmplete protease inhibitor cocktail (Roche)) and incubated for 10 min at 4 °C with slight agitation. After centrifugation, the cell pellet was resuspended in 3× PV Buffer A containing 0.1% NP-40 and incubated for 10 min at 4 °C with slight agitation. Nuclei were recovered by centrifugation and the soluble nuclear fraction was extracted for 1 h at 4 °C with slight agitation using 1× PV Buffer C (10 mM Hepes pH 7.9, 400 mM NaCl, 1.5 mM MgCl2, 26% glycerol, 0.2 mM EDTA, cOmplete protease inhibitor cocktail). Protein concentration was measured using the Bradford assay (BioRad).

Nuclear extract samples were mixed with 1× SDS loading buffer (2% SDS, 0.1 M Tris pH 6.8, 0.1 M DTT, 10% glycerol, 0.1% bromophenol blue) and placed at 95 °C for 5 min. Between 25–35 μg of nuclear extract was separated on home-made SDS-PAGE gels or NuPAGE 3–8% Tris-acetate gels (Life Technologies, for large Mediator subunits). Gels were blotted onto nitrocellulose membranes using the Trans-Blot Turbo transfer system (BioRad). Antibodies used for Western blot analysis were rabbit polyclonal anti-MED13L (A302-420A, Bethyl laboratories), rabbit polyclonal anti-MED13 (GTX129674, Genetex), rabbit monoclonal anti-CDK8 (ab229192, Abcam), rabbit polyclonal anti-CCNC (A301-989A, Bethyl laboratories), rabbit polyclonal anti-MED1 (A300-793A, Bethyl laboratories), rabbit polyclonal anti-MED15 (A302-422A, Bethyl laboratories), rabbit polyclonal anti-MED23 (A300-425A, Bethyl laboratories), rabbit polyclonal anti-MED17 (GTX115241, Genetex), rabbit polyclonal anti-MED14 (A301-044A-T, Bethyl laboratories), rabbit monoclonal anti-RING1B (5694, Cell Signaling), rabbit monoclonal anti-SUZ12 (3737, Cell Signaling), rabbit polyclonal anti-PCGF2 (sc-10744, Santa Cruz), rabbit monoclonal anti-T7-Tag (D9E1X, 13246, Cell Signaling), mouse monoclonal anti-TBP (ab818, Abcam), rabbit monoclonal anti-HDAC1 (ab109411, Abcam), and mouse monoclonal anti-Flag (F1804, Sigma). Images were analyzed using Image Studio v5.2 (LI-COR).

### Co-immunoprecipitation of the CKM−Mediator complex

For purification of the CKM−Mediator complex from wild type or tamoxifen-treated *Med13/13l*ᶠˡ/ᶠˡ ESCs, 600 μg of nuclear extract was diluted in BC150 buffer (50 mM Hepes pH 7.9, 150 mM KCl, 0.5 mM EDTA, 0.5 mM DTT, cOmplete protease inhibitor cocktail (Roche)). Samples were incubated with 5 μg CDK8 antibody (A302-500A, Bethyl laboratories) and 25 units benzonase nuclease (Millipore) overnight at 4 °C. For purification of T7-MED14, 5 μl T7-Tag antibody (D9E1X, 13246, Cell Signaling) and 25 units benzonase nuclease were used. Protein A agarose beads (RepliGen) were blocked for 1 h at 4 °C in Buffer BC150 containing 1% fish skin gelatin (Sigma) and 0.2 mg ml⁻¹ BSA (New England Biolabs). The blocked beads were added to the samples and incubated for 4 h at 4 °C. Four washes for 10 min each were performed using BC150 containing 0.02% NP-40. The beads were resuspended in 2× SDS loading buffer and boiled for 5 min to elute the immunoprecipitated complexes.

### Chromatin immunoprecipitation

Chromatin immunoprecipitation was performed as described previously[55]. In brief, $50 \times 10^6$ ES cells were fixed for 45 min with 2 mM DSG (Thermo Fisher Scientific) in PBS followed by 12.5 min with 1% formaldehyde (methanol-free, Thermo Fisher Scientific). Reactions were quenched by the addition of glycine to a final concentration of 125 μM and the fixed cells were washed in ice-cold PBS and snap frozen in liquid nitrogen. $50 \times 10^6$ HEK293T cells were fixed as above, snap frozen in $2 \times 10^6$ aliquots and stored at −80 °C until further use.

For calibrated ChIP−seq, $2 \times 10^6$ HEK293T cells were resuspended in 1 ml ice-cold lysis buffer (50 mM HEPES pH 7.9, 140 mM NaCl, 1 mM EDTA, 10% glycerol, 0.5% NP-40, 0.25% Triton X-100) and added to $50 \times 10^6$ fixed ESCs resuspended in 9 ml lysis buffer. The cell suspension was incubated for 10 min at 4 °C. The released nuclei were washed (10 mM Tris-HCl pH 8, 200 mM NaCl, 1 mM EDTA, 0.5 mM EGTA) for 10 min at 4 °C. The chromatin pellet was resuspended in 1 ml of ice-cold sonication buffer (10 mM Tris-HCl pH 8, 100 mM NaCl, 1 mM EDTA, 0.5 mM EGTA, 0.1% Na deoxycholate, 0.5% N-lauroylsarcosine) and sonicated for 25 cycles (30 s on and 30 s off) using a BioRuptor Pico sonicator (Diagenode), shearing genomic DNA to produce fragments between 300 bp and 1 kb. Following sonication, Triton X-100 was added to a final concentration of 1%. Two hundred and fifty μg chromatin was diluted ten-fold in ChIP dilution buffer (1% Triton X-100, 1 mM EDTA, 20 mM Tris-HCl pH 8.0, 150 mM NaCl) and used in each immunoprecipitation. Three reactions per treatment condition were set up to allow

for maximal DNA recovery suitable for library preparation. Chromatin was pre-cleared with protein A Dynabeads (Thermo Fisher Scientific), blocked with 0.2 mg ml[-1] BSA and 50 μg ml[-1] yeast tRNA and incubated with the respective antibodies overnight at 4 °C. Antibody-bound chromatin was purified using blocked protein A Dynabeads for 3 h at 4 °C. ChIP washes were performed as described previously[89]. ChIP DNA was eluted in ChIP elution buffer (1% SDS, 100 mM NaHCO3) and reverse crosslinked overnight at 65 °C with 200 mM NaCl and RNase A (Sigma). The reverse crosslinked samples were treated with 20 μg ml[-1] proteinase K and purified using a ChIP DNA Clean and Concentrator kit (Zymo Research). The three reactions per treatment condition were pooled at this stage. For each sample, corresponding input DNA was also reverse crosslinked and purified. The efficiency of the ChIP experiments was confirmed by quantitative PCR. Prior to library preparation, 5–10 ng ChIP material was diluted to 50 μl in TLE buffer (10 mM Tris-HCl pH 8.0, 0.1 mM EDTA) and sonicated with a Bioruptor Pico sonicator for 17 min (30 s on and 30 s off).

The antibodies used for ChIP–seq experiments were rabbit polyclonal anti-CDK8 (A302-500A, Bethyl laboratories, 2.5 μl), rabbit monoclonal anti-RING1B (5694, Cell Signaling, 3 μl), rabbit polyclonal anti-PCGF2 (sc-10744, Santa Cruz, 3 μl), rabbit polyclonal anti-CBX7 (ab21873, abcam, 4 μl), rabbit monoclonal anti-T7-Tag (D9E1X) (13246, Cell Signaling, 3 μl). The antibodies used for ChIP-quantitative PCR for TetO targeting experiments were rabbit polyclonal anti-FS2 (produced in-house[89], 33 μl), polyclonal anti-MED12 (A300-774A, Bethyl laboratories, 3 μl), polyclonal anti-MED1 (A300-793A, Bethyl laboratories, 3 μl), polyclonal anti-CCNC (A301-989A, Bethyl laboratories, 3 μl) and rabbit polyclonal anti-FS2 (produced in-house, 5 μl).

### Native chromatin immunoprecipitation

Native calibrated ChIP–seq for H3K27me3 was performed as described previously[59,89]. In brief, 50 × 10^6 ESCs were mixed with 20 × 10^6 SG4 Drosophila cells and washed with 1× PBS prior to chromatin isolation. Nuclei were released in ice-cold lysis buffer (10 mM Tris-HCl pH 8.0, 10 mM NaCl, 3 mM MgCl2, 0.1% NP-40), washed and resuspended in 1 ml ice-cold digestion buffer (10 mM Tris-HCl pH 8.0, 10 mM NaCl, 3 mM MgCl2, 0.1% NP-40, 0.25 M sucrose, 3 mM CaCl2, 1× cOmplete protease inhibitor cocktail (Roche)). Chromatin was digested with 200 units MNase (Thermo Fisher Scientific) for 5 min at 37 °C, and the reaction was stopped by the addition of 4 mM EDTA pH 8.0. The samples were centrifuged at 1,500×g for 5 min at 4 °C, the supernatant (S1) was retained. The remaining pellet was incubated with 300 μl of nucleosome release buffer (10 mM Tris-HCl pH 7.5, 10 mM NaCl, 0.2 mM EDTA, 1× protease inhibitor cocktail (Roche)) at 4 °C for 1 h, passed five times through a 27 guage needle using a 1 mL syringe, and spun at 1,500×g for 5 min at 4 °C. The second supernatant (S2) was collected and combined with the corresponding S1 sample from above. Digestion to mostly mononucleosomes was confirmed on a 1.5% agarose gel. The prepared native chromatin was aliquoted, snap frozen in liquid nitrogen, and stored at −80 °C until further use. ChIPs were performed as described previously[59], using 5 μl of H3K27me3 antibody prepared in-house.

### Calibrated nuclear RNA-seq

Nuclear RNA sample preparation was performed using 20 × 10^6 ES or RA-treated cells and 8 × 10^6 SG4 Drosophila cells, as described previously[59]. RNA was isolated from purified nuclei using a RNeasy RNA extraction kit (Qiagen), and genomic DNA contamination was depleted using a TURBO DNA-free Kit (Thermo Fisher Scientific). The quality of RNA was assessed using a 2100 Bioanalyzer RNA 6000 Pico kit (Agilent). All cnRNA-seq experiments were performed in biological quadruplicates.

### Library preparation and high-throughput sequencing

All cChIP–seq experiments were performed in biological triplicates. All ncRNA-seq experiments were performed in biological quadruplicates.

Libraries for cChIP–seq and native cChIP–seq were prepared from 5–10 ng of ChIP and corresponding input DNA samples using a NEBNext Ultra II DNA Library Prep Kit for Illumina (New England Biolabs), following the manufacturer's guidelines. For ncRNA-seq, RNA samples (800 ng) were depleted of ribosomal RNA using the NEBNext rRNA Depletion kit (New England Biolabs). RNA-seq libraries were prepared using the NEBNext Ultra Directional RNA Library Prep kit (New England Biolabs). Samples were indexed using NEBNext Multiplex Oligos (New England Biolabs). The average size and concentration of all libraries were analyzed using the 2100 Bioanalyzer High Sensitivity DNA Kit (Agilent) followed by qPCR using SensiMix SYBR (Bioline, UK) and KAPA Illumina DNA standards (Roche). Libraries were sequenced as 40 bp paired-end reads on Illumina NextSeq 500 platform.

### Massively parallel sequencing, data processing and normalization

For cChIP–seq, paired-end reads were aligned to concatenated mouse and spike-in genomes (mm10 + hg19 for crosslinked cChIP–seq and mm10 + dm6 for native cChIP–seq) using Bowtie 2 (ref. [90]) with the '–no-mixed' and '–no-discordant' options specified. Reads that were mapped more than once were discarded, followed by removal of PCR duplicates using Sambamba[91].

For cnRNA-seq, paired-end reads were first aligned using Bowtie 2 (with '–very-fast,' '–no-mixed' and '–no-discordant' options) against the concatenated mm10 and dm6 rRNA genomic sequences (GenBank: BK000964.3 and M21017.1) to filter out reads mapping to ribosomal RNA gene fragments. All unmapped reads were then aligned against the genome sequence of concatenated mm10 and dm6 genomes using STAR[92]. To improve mapping of intronic sequences of nascent transcripts abundant in nuclear RNA-seq, reads failing to map using STAR were aligned against the mm10 + dm6 concatenated genome using Bowtie 2 (with '–sensitive-local,' '–no-mixed' and '–no-discordant' options). PCR duplicates were removed using SAMTools[93].

For visualization and annotation of genomic regions, internal normalization of cChIP–seq and ncRNA-seq experiments was performed as described previously[59]. In brief, mouse reads were randomly downsampled based on the spike-in ratio (hg19 or dm6) in each sample. To account for possible spike-in cell variation, the ratio of spike-in to mouse read counts in the corresponding ChIP inputs were used as correction factors for cChIP–seq replicates. MED14-T7 ChIP–seq was performed without spike-in normalisation. Individual replicates were compared using multiBamSummary and plotCorrelation functions from deepTools (version 3.1.1)[94], confirming a high degree of correlation (Pearson's correlation coefficient > 0.9). Replicates were pooled for downstream analysis. Genome-coverage tracks for visualization on the UCSC genome browser[95] were generated using the pileup function from MACS2 (ref. [96]) for ChIP–seq and genomeCoverageBed from BEDtools (v2.17.0) (ref. [97]) for cnRNA-seq.

### Read count quantification and analysis

Heatmap and metaplot analysis for ChIP–seq was performed using computeMatrix and plotProfile and plotHeatmap functions from deepTools (v.3.1.1)[94], looking at read density at Polycomb domains, CDK8 peaks or transcription start sties (TSSs) of CKM–MED-dependent genes. Intervals of interest were annotated with read counts from merged replicates, using a custom-made Perl script utilising SAMtools (v1.7) (ref. [93]). Polycomb domains were defined in ref. [59]. CDK8 peaks were defined in ref. [55]. H3K27me3ac peaks were defined in ref. [44].

For differential gene expression analysis, read counts were obtained from the non-normalized mm10 BAM files for a non-redundant mouse gene set, using a custom-made Perl script utilizing SAMtools (v1.7) (ref. [93]). The non-redundant mouse gene set (n = 20,633) was obtained by filtering mm10 refGenes for very short genes with poor sequence mappability and highly similar transcripts. To identify significant changes in gene expression, a custom-made R script utilizing

DESeq2 (ref. [98]) was used. For spike-in normalization, read counts for the spike-in genome at a unique set of dm6 refGenes were supplied to calculate DESeq2 size factors which were then used for DESeq2 normalization of raw mm10 read counts, similarly to ref. [99]. For a change to be considered significant, a threshold fold change of > 1.5 and adjusted $P < 0.05$ was applied.

The distribution of $\log_2$ fold changes and normalized read counts at different genomics intervals was visualized using custom R scripts. For boxplot analyses, boxes showing interquartile range (IQR) and whiskers extending by no more than $1.5 \times$ IQR were used.

## Hi-C library preparation and analysis

In situ Hi-C in *Med13/13l*[fl/fl] ESCs was performed in biological duplicates as described in ref. [100]. Hi-C libraries were sequenced on the Illumina NextSeq 500 platform as 51 bp or 40 bp paired-end reads. Hi-C sequencing data were mapped to GRCm38.p6 and processed with Hi-C-Pro 2.9 (ref. [101]). Further data analysis was performed with GENOVA (http://www.github.com/deWitLab/GENOVA)[102].

TAD and loop coordinates of mouse ESC samples were taken from ref. [25]. Aggregate peak analysis (APA) and aggregate TAD analysis (ATA) were performed on 10 kb ice-normalized matrices with default parameters. Paired-end spatial chromatin analysis (PE-SCAn) between the 100 kb regions surrounding Ring1B peaks in Polycomb domains was also performed on these matrices. Super-enhancer coordinates for GRCm38.p6 were downloaded from dbSUPER[103]. PE-SCAn between the 1 Mb regions surrounding super enhancers was performed using 20 kb ice-normalized matrices, setting the top and bottom 5% values as outliers.

## Capture-C extraction protocol

Chromatin was extracted and fixed as described previously[104]. In brief, $10 \times 10^6$ mouse ESCs were trypsinized, collected in 50 ml falcon tubes in 9.3 ml medium, and crosslinked with 1.25 ml 16% formaldehyde (1.89% final concentration; methanol-free, Thermo Fisher Scientific) while rotating for 10 min at 25 °C. Cells were quenched with 1.5 ml 1 M cold glycine, washed with cold PBS and lysed for 20 min at 4 °C in lysis buffer (10 mM Tris pH 8, 10 mM NaCl, 0.2% NP-40, supplemented with cOmplete proteinase inhibitors (Roche)) prior to snap freezing in 1 ml lysis buffer on dry ice. Fixed chromatin was stored at −80 °C.

## Capture-C library construction protocol

Capture-C libraries were prepared as described previously[105]. In brief, lysates were thawed on ice, pelleted and resuspended in 650 μl 1× DpnII buffer (New England Biolabs). Three 1.5 ml tubes with 200 μl lysate each were treated in parallel with 0.28% final concentration of SDS (Thermo Fisher Scientific) for 1 h at 37 °C in a thermomixer shaking at 500 r.p.m. (30 s on/off). Reactions were quenched with 1.67% final concentration of Triton X-100 for 1 h at 37 °C in a thermomixer shaking at 500 r.p.m. (30 s on/off) and digested for 24 h with 3 × 10 μl DpnII (produced in-house) at 37 °C in a thermomixer shaking at 500 r.p.m. (30 s on/off). An aliquot from each reaction (100 μl) was taken for use as the digestion control, reverse crosslinked and visualized on an agarose gel. The remaining chromatin was then independently ligated with 8 μl T4 Ligase (240 units Thermo Fisher Scientific) in a volume of 1440 μl for 20 h at 16 °C. The nuclei containing ligated chromatin were pelleted to remove any non-nuclear chromatin and reverse crosslinked, and the ligated DNA was phenol-chloroform purified. The sample was resuspended in 300 μl water and sonicated for 13 cycles (30 s on/off) using a Bioruptor Pico (Diagenode) to achieve a fragment size of approximately 200 bp. Fragments were size-selected using AmpureX beads (Beckman Coulter) and a 0.85×/0.4× selection ratio. Two reactions of 1–5 μg DNA each were adapter-ligated and indexed using the NEBNext Ultra II DNA Library Prep Kit for Illumina (New England Biolabs) and NEBNext Multiplex Oligos for Illumina Primer sets 1 and 2 (New England Biolabs). The libraries were amplified

with seven PCR cycles using the Herculase II Fusion Polymerase kit (Agilent). Libraries were hybridized in the following way: for each promoter containing a DpnII restriction fragment, we designed two 70 bp capture probes using the CapSequm online tool (http://apps.molbiol.ox.ac.uk/CaptureC/cgi-bin/CapSequm.cgi) with the following filtering parameters: duplicates, < 2; density, < 30; SRepeatLength, < 30; duplication, FALSE. For promoters for which no probes could be designed for the restriction fragment directly overlapping the TSS, probes were designed for the next-nearest DpnII fragment, if it was within 500 bp of the TSS. The probes were pooled at 2.9 nM each, and the samples were multiplexed en masse prior to hybridization (2 μg each, according to Qubit dsDNA BR Assay, Invitrogen). Hybridization was carried out using the Nimblegen SeqCap system (Roche, Nimblegen SeqCap EZ HE-oligo kit A no. 6777287001, Nimblegen SeqCap EZ HE-oligo kit B no 06777317001, Nimblegen SeqCap EZ Accessory kit v2 no. 07145594001, Nimblegen SeqCap EZ Hybridization and wash kit no. 05634261001), according to the Roche protocol, for 72 h followed by a 24 h hybridization (double capture). Captured libraries were quantified by qPCR using SensiMix SYBR (Bioline) and KAPA Illumina DNA standards (Roche) and sequenced on an Illumina NextSeq 500 as 40 bp paired-end reads. Libraries for Capture-C in *Med13/13l*[fl/fl] and *Pcgf4*[−/−] *Pcgf2*[fl/fl] were generated using biological triplicates (Capture set1) or biological duplicates (Capture set2, as control for captures in the TetR-fusion lines). Libraries for Capture-C in the TetR-fusion lines were generated in biological triplicates.

## Capture-C analysis

Paired-end reads were aligned to mm10 (or mm10 + BAC insert for TetR-fusion cell lines) and filtered for Hi-C artifacts using HiCUP[106] and Bowtie 2 (ref. [90]), with the fragment filter set to 100–800 bp. Read counts of reads aligning to captured gene promoters and interaction scores (=significant interactions) were then called by CHiCAGO[107].

For visualisation of Capture-C data, weighted, pooled read counts from CHiCAGO data files were normalized to total read counts aligning to captured gene promoters in the sample and then to the number of promoters in the respective capture experiment and multiplied by a constant number to simplify genome browser visualization using the following formula: normCounts=1/cov*nprom*100000. Bigwig files were generated from these normalized read counts.

For comparative boxplot analysis, we first determined all interactions between promoters and a given set of intervals (that is, Polycomb domains) using a CHiCAGO score of ≥5 as a cutoff. Next, for each promoter-interval interaction, we quantified the sum of normalized read counts or CHiCAGO scores across all DpnII fragments overlapping this interval. This number was then divided by the total number of interval-overlapping DpnII fragments to obtain mean normalized read counts and scores. For boxplot analyses, boxes show IQR and whiskers show the most extreme data point, which is no more than by $1.5 \times$ IQR.

## Statistics and reproducibility

Details of the individual statistical analyses and tests, as well as the number of biological replicates, can be found in the respective figure legends and in the detailed methods description. No statistical method was used to predetermine sample size. No data were excluded from the analyses. The experiments were not randomized. The investigators were not blinded to allocation during experiments and outcome assessment.

## Reporting summary

Further information on research design is available in the Nature Research Reporting Summary linked to this article.

## Data availability

The datasets generated in this study are available from GEO database under accession number GSE185930. Published data used in this study

include mouse ESC TAD and loop coordinates (GSE96107 (ref. [25])), Polycomb domains (GSE119620 (ref. [59])), CDK8 peaks (GSE98756 (ref. [55])) and H3K27Ac peaks (GSE136424 (ref. [44])). For cnRNA-seq processing, we used mm10 (GenBank: BK000964.3) and dm6 (GenBank: M21017.1) rDNA genomic datasets. Source data are provided with this paper.

## Code availability

Custom R and Perl scripts used for cCHIP–seq and cnRNA-seq data analysis in this study have been developed previously[108] and are available at https://github.com/nFursova/Calibrated_ChIPseq_RNAseq. All R scripts for Capture-C analysis are available upon request. GENOVA is an open source software package available at http://www.github.com/deWitLab/GENOVA.

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

## Acknowledgements

We would like to thank A. Hughes, N. Fursova and N. Blackledge for critical reading of the manuscript, as well as members of the Klose Lab for fruitful discussions. We thank N. Fursova for help with DESeq2 analysis. We thank T. Milne and J. Hughes for sharing TetO Capture-C probe sequences. We are grateful to A. Williams at the Department of Zoology, University of Oxford, for sequencing support on the NextSeq 500. Work in the Klose Lab is supported by the Wellcome Trust (209400/Z/17/Z) and the European Research Council (681440). A.F. is supported by a Sir Henry Wellcome Postdoctoral Fellowship. Work in the de Wit laboratory is supported by an ERC StG 637587 (HAP-PHEN) and a Vidi grant from the Netherlands Scientific Organization (NWO, 016.16.316).

## Author contributions

E.D. and R.J.K. conceived the project and wrote the manuscript with contributions from all co-authors. E.D. performed experiments, data analysis and visualization. A.F. and E.D. generated Capture-C libraries and A.F. performed Capture-C data analysis. K.D.F. generated Hi-C libraries and R.H.W. performed Hi-C data analysis. A.L. generated the MED14-T7 *Med13/13l*<sup>fl/fl</sup> ESC line and libraries. E.W. and R.J.K. supervised the project.

## Competing interests

The authors declare no competing interests.

## Additional information

**Extended data** is available for this paper at https://doi.org/10.1038/s41594-022-00840-5.

**Correspondence and requests for materials** should be addressed to Emilia Dimitrova or Robert J. Klose.

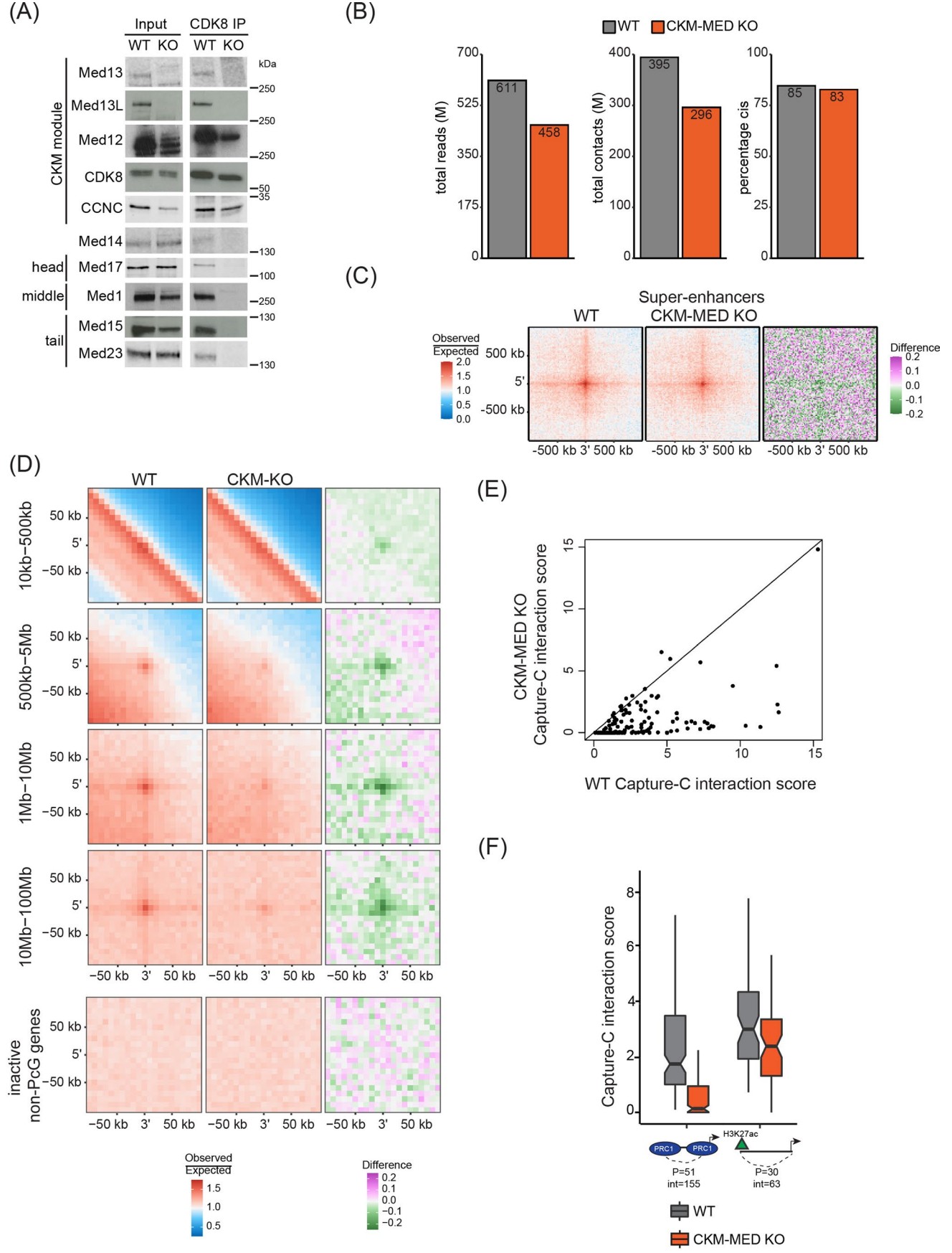

**Extended Data Fig. 1 | See next page for caption.**

**Extended Data Fig. 1 | CKM-Mediator has a limited role in 3D genome organisation but is essential for Polycomb domain interactions. a,** A representative Western blot analysis of CDK8 immunoprecipitation (n = 2) from nuclear extracts from *Med13/13l*$^{fl/fl}$ (WT) and *Med13/13l*$^{-/-}$ (CKM-Mediator KO) ESCs, probed with the indicated antibodies. **b,** Quality control metrics of the Hi-C data, showing total sequenced read-pairs in millions, total valid contacts in millions and percentages *in cis* contacts for WT and CKM-Mediator KO ESCs. **c,** Aggregate analysis of super enhancer interactions in WT and CKM-Mediator KO ESCs. The difference between WT and KO is shown. **d,** Aggregate analysis of Hi-C signal (10 kb resolution) at pairs of Polycomb domains at the indicated distance ranges in *Med13/13l*$^{fl/fl}$ (WT) and *Med13/13l*$^{-/-}$ (CKM-Mediator KO) ESCs, with 200 kb flanking regions. Interactions of inactive non-Polycomb gene promoters subsampled to match regions as in Fig. 1f (n = 2096), are included as a negative control (bottom). The difference between WT and KO is shown. **e,** Capture-C interaction scores for interactions between Polycomb domains in WT and CKM-Mediator KO ESCs (number of promoters = 51, number of interactions = 148). **f,** Boxplot analysis of Capture-C interaction scores from WT and CKM-Mediator KO ESCs showing interactions between Polycomb gene promoters and other Polycomb-domains (left), or non-Polycomb gene promoters with active sites (H3K27ac, right). Number of promoters (P) and interactions (int) is shown. Boxes show interquartile range, center line represents median, whiskers extend by 1.5x IQR or the most extreme point (whichever is closer to the median), while notches extend by 1.58x IQR/sqrt(n), giving a roughly 95% confidence interval for comparing medians.

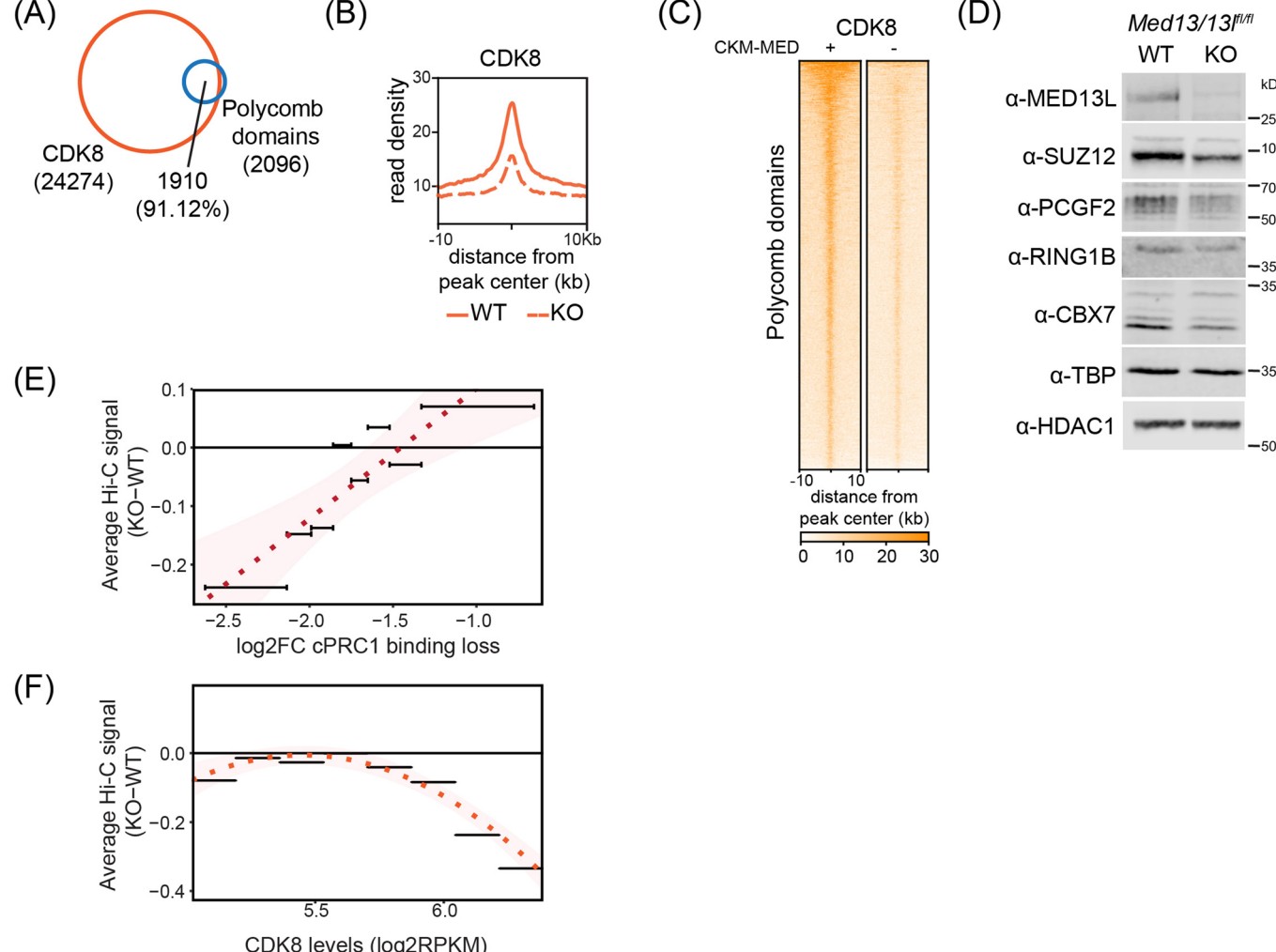

**Extended Data Fig. 2 | CKM-Mediator regulates canonical PRC1 binding. a**, A Venn diagram showing the overlap between CDK8 peaks and Polycomb domains. Number of peaks and percent overlap are indicated. **b,** Metaplot analysis of CDK8 enrichment at Polycomb domains (n = 2097) in WT and CKM-Mediator (CKM-MED) KO ESCs. **c**, Heatmaps showing CDK8 ChIPseq signal at Polycomb domains (n = 2097) in WT and CKM-Mediator KO ESCs, sorted by decreasing RING1B signal. **d,** A representative Western blot analysis (n = 6) of nuclear extracts from WT and CKM-MED KO ESCs probed with the indicated antibodies. TBP and HDAC1 are used as loading controls. **e**, Comparison between loss of Hi-C signal (difference between CKM-MED-KO and WT) and loss of cPRC1 (PCGF2) binding (log2 fold change) at Polycomb domains. Polycomb domains were divided into equal bins (261 domains each) based on log2 fold change in cPRC1 binding. **f,** Comparison between loss of Hi-C signal (difference between CKM-MED-KO and WT) and levels of CDK8 binding in WT cells (log2RPKM). Domains were divided into eight bins based on CDK8 RPKM levels.

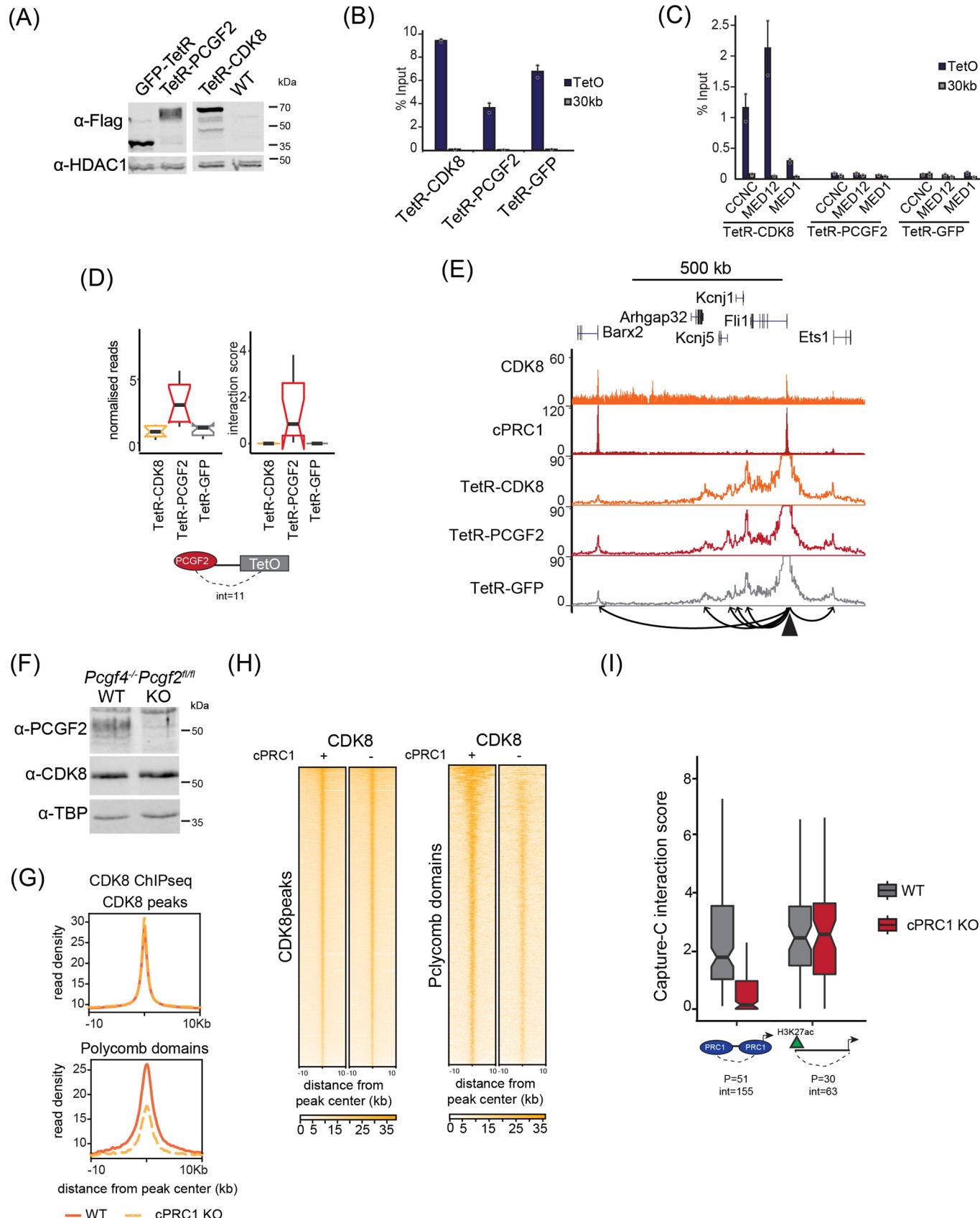

**Extended Data Fig. 3 | See next page for caption.**

**Extended Data Fig. 3 | cPRC1 creates interactions between Polycomb domains. a**, A representative Western blot analysis (n = 3) of nuclear extracts from the TetR-fusion lines used for Capture-C analysis probed with anti-Flag antibody to detect expression of the fusion proteins. HDAC1 is used as a loading control. **b**, ChIP-qPCR analysis of binding of the different TetR-fusion lines to the TetO array. Data are presented as mean value (n = 2) ±SD. Data points for individual replicates are shown. **c,** ChIP-qPCR analysis of binding of the CKM-Mediator complex to the TetO array in the TetR-CDK8, TetR-PCGF2, and TetR-GFP lines.. Data are presented as mean value (n = 2 for TetR-CDK8 and n = 3 for TetR-PCGF2 and TetR-GFP) ± SD. Data points for individual replicates are shown. **d**, Boxplot analysis of Capture-C mean normalised read counts and interaction scores in the TetR-fusion lines, looking at interactions with Polycomb domains (PCGF2-bound). Number of interactions is shown. Boxes show interquartile range, center line represents median, whiskers extend by 1.5x IQR or the most extreme point (whichever is closer to the median), while notches extend by 1.58x IQR/sqrt(n), giving a roughly 95% confidence interval for comparing medians. **e**, Snapshots showing Capture-C read count signal from TetR-CDK8, TetR-PCGF2

and TetR-GFP lines at a control locus. CDK8 and PCGF2 (cPRC1) ChIPseq signal is given as a reference. The *Fli1* promoter bait is shown as a triangle and interactions created with surrounding cPRC1-bound sites are represented with arrowheads. **f**, A representative Western blot analysis of nuclear extracts (n = 3) from WT and cPRC1 KO ESCs probed with the indicated antibodies. TBP is used as a loading control. **g**, Metaplot analysis of CDK8 enrichment at CDK8 peaks (n = 24275) and Polycomb domains (n = 2097) in WT and cPRC1 KO ESCs. **h**, Heatmaps showing CDK8 ChIPseq signal at CDK8 peaks (n = 24275) and Polycomb domains (n = 2097) in WT and cPRC1 KO ESCs, sorted by decreasing CDK8 or RING1B signal, respectively. **i,** Boxplot analysis of Capture-C interaction scores from WT and cPRC1 KO ESCs showing interactions between Polycomb gene promoters and other Polycomb-domains (left), or non-Polycomb gene promoters and active sites (H3K27ac, right). Number of promoters (P) and interactions (int) is shown. Boxes show interquartile range, center line represents median, whiskers extend by 1.5x IQR or the most extreme point (whichever is closer to the median), while notches extend by 1.58x IQR/sqrt(n), giving a roughly 95% confidence interval for comparing medians.

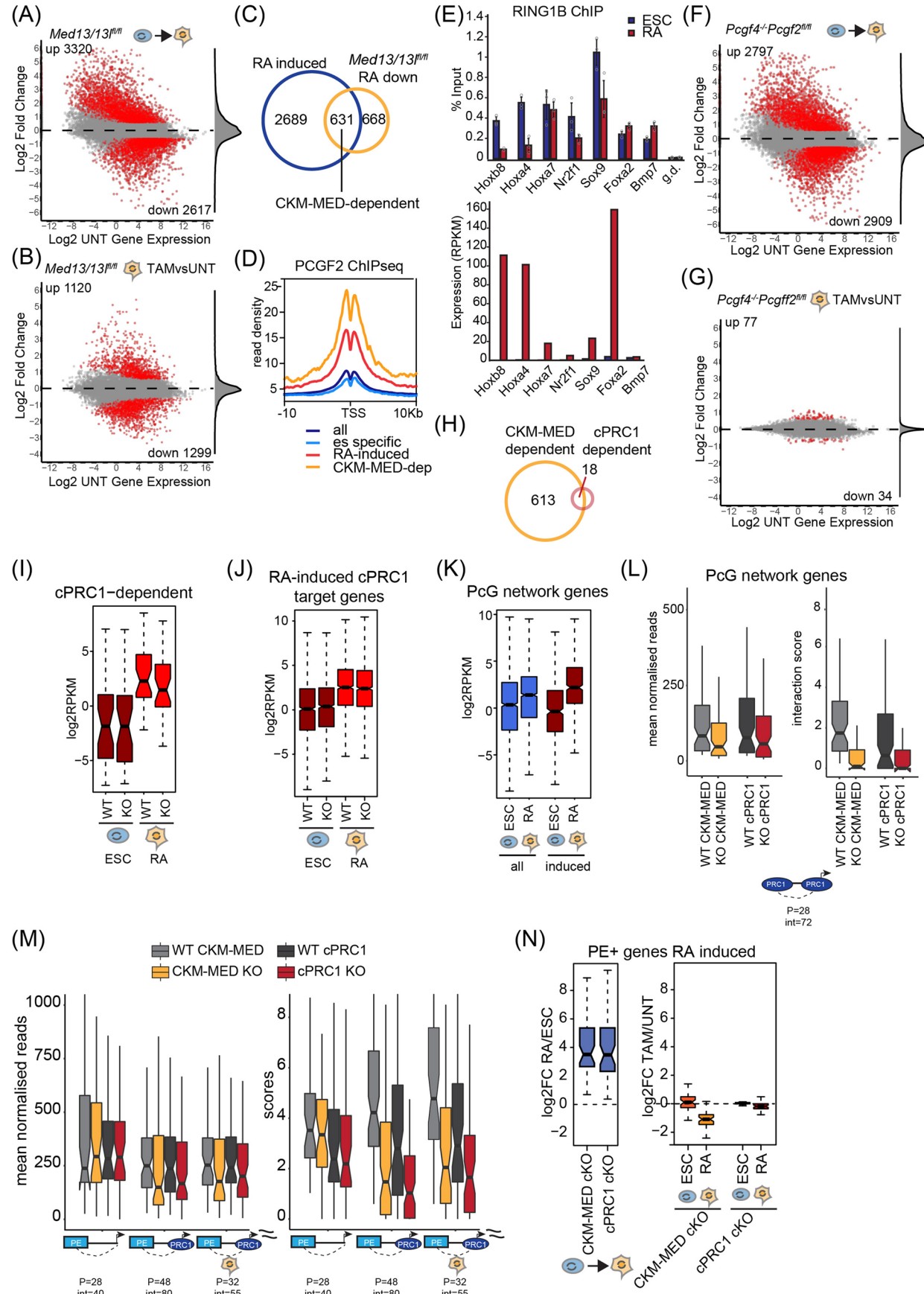

**Extended Data Fig. 4 | See next page for caption.**

**Extended Data Fig. 4 | CKM-Mediator primes genes for activation during differentiation independently of cPRC1-mediated interactions. a,** An MA plot of log2 fold changes in gene expression (cnRNA-seq) WT ESCs and RA-treated cells, determined using DESeq2. Significant expression changes (>1.5 fold change and padj<0.05) are shown in red and number of genes is indicated. Distribution of gene expression changes is shown on the right as a density. **b,** An MA plot of log2 fold changes in gene expression (cnRNAseq) WT and CKM-Mediator KO RA-treated cells, determined using DESeq2. Significant expression changes (>1.5 fold change and padj<0.05) are shown in red and number of genes is indicated. Distribution of gene expression changes is shown on the right as a density. **c,** A Venn diagram showing the overlap between RA-induced genes as defined in **a** and genes downregulated in CKM-Mediator KO cells, following RA treatment, as defined in **b. d,** A metaplot showing enrichment of cPRC1 (PCGF2) over the transcription start site (TSS) of the indicated different classes of genes. All=20633, ES-specific=2617; RA-induced=3320; CKM-Mediator-dependent=631. **e,** ChIP-qPCR showing enrichment of RING1B at promoters of developmental genes in WT ESCs and RA-induced cells (top). Data are presented as mean value (n = 3) ±SD. Data points for individual replicates are shown. Gene desert (g.d.) is included as a negative control region. Expression of the corresponding genes (RPKM) is shown below. Bpm7 is a control, non-induced gene. **f,** As in **a** for cPRC1 cKO cells. **g,** As in **b** for cPRC1 cKO cells. **h,** A Venn diagram showing the overlap between CKM-Mediator-dependent and cPRC1-dependent genes. Gene numbers are indicated. **i,** Boxplot analysis of the expression of cPRC1-dependent genes

(n = 34), as defined in Extended Data Fig. 4f. Boxes show interquartile range, center line represents median, whiskers extend by 1.5x IQR or the most extreme point (whichever is closer to the median), while notches extend by 1.58x IQR/sqrt(n), giving a roughly 95% confidence interval for comparing medians. **j,** Boxplot analysis of the expression of RA-induced cPRC1 (PCGF2) target genes (n = 1201) in WT and cPRC1 KO ESCs and following RA-induction. Boxes are defined as in **i. k,** Boxplot analysis of the expression of genes within the Polycomb network in ESCs and following RA induction (all = 1974; RA-induced=482). Boxes are defined as in **i. i,** Boxplot analysis of Capture-C mean normalised read counts (left) and interaction score (right) from CKM-Mediator cKO and cPRC1 cKO ESCs showing interactions between promoters of genes within the Polycomb (PcG) network and Polycomb domains. Number of promoters (p) and interactions (int) is shown. Boxes are defined as in **i. m,** Boxplot analysis of Capture-C mean normalised read counts (left) and interaction score (right) from CKM-Mediator cKO and cPRC1 cKO ESCs showing interactions between gene promoters and poised enhancers (PE). Genes were divided into non-Polycomb targets (left set), Polycomb targets (middle set) and Polycomb targets induced by RA (right set). Number of promoters (P) and interactions (int) is shown. Boxes are defined as in **i. n,** Boxplot analysis of the expression of RA-induced genes that interact with a poised enhancer (n = 55) in CKM-Mediator cKO and cPRC1 cKO cells. The difference between WT RA cells and ESCs (left), as well as KO and WT cells (right), is shown as log2FC. Boxes are defined as in **i.**

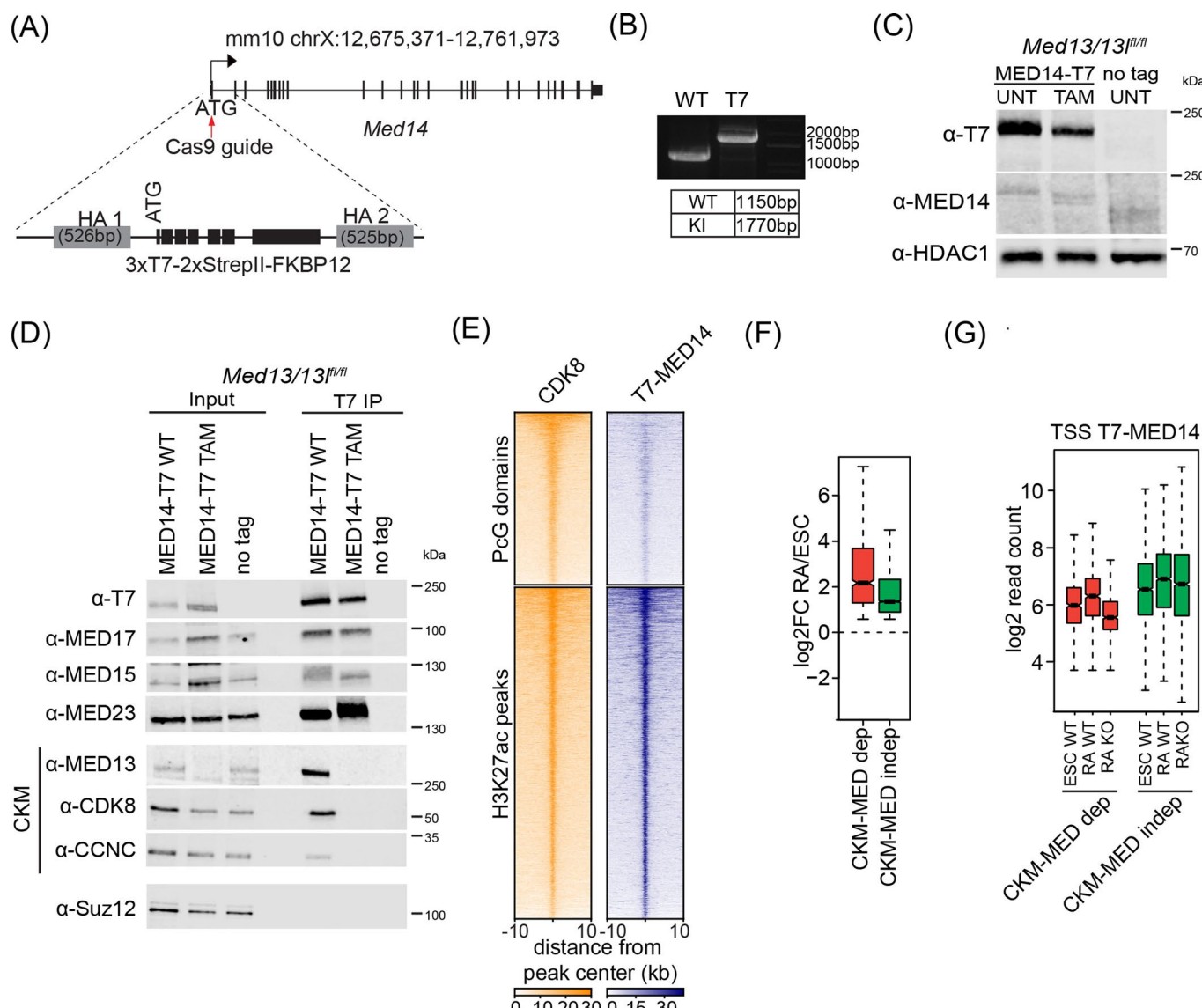

**Extended Data Fig. 5 | CKM-Mediator enables gene induction via recruitment of the Mediator complex. a,** A schematic illustration of the generation of the T7-MED14 expressing *Med13/13l^fl/fl^* ESC line. **b,** PCR showing amplification of homozygously-tagged T7-Med14 alleles (n = 2). **c,** A representative Western blot analysis of nuclear extracts from the T7-MED14 *Med13/13l^fl/fl^* ESC line, following tamoxifen (TAM) treatment (n = 3). Extract from an untagged ESC line was used as a control. HDAC1 was used as a loading control. **d,** A representative immunoprecipitation (IP) of endogenously T7-tagged MED14 with T7 antibody using nuclear extracts from *Med13/13l^fl/f^* ESCs before (UNT) and after tamoxifen (TAM) treatment (n = 2). The IPs were probed with the indicated antibodies. An IP from an untagged ESC line was performed as a negative control and a Western blot for SUZ12 was included as a control protein that does not interact

with Mediator. **e,** Heatmaps of CDK8 and T7-MED14 ChIPseq signal at Polycomb domains (n = 2097) and H3K27ac peaks (n = 4037), sorted by decreasing CDK8 signal. **f,** Boxplots showing gene expression change (log2FC) of CKM-Mediator-dependent (n = 631) and CKM-Mediator-independent (n = 2689) RA-induced genes following RA differentiation of WT ESCs. Boxes show interquartile range, center line represents median, whiskers extend by 1.5x IQR or the most extreme point (whichever is closer to the median), while notches extend by 1.58x IQR/ sqrt(n), giving a roughly 95% confidence interval for comparing medians. **g,** Boxplots showing T7-MED14 ChIPseq signal at the TSS (1000 bp) of the different classes of RA-induced gene classes as defined in **e** in ESCs and RA-induced cells (WT and CKM-Mediator KO). Boxes are defined as in **f**. Signal is an average from three independent biological experiments.

# Reporting Summary

## Statistics

For all statistical analyses, confirm that the following items are present in the figure legend, table legend, main text, or Methods section.

| n/a | Confirmed | |
|---|---|---|
| ☐ | ☒ | The exact sample size (*n*) for each experimental group/condition, given as a discrete number and unit of measurement |
| ☐ | ☒ | A statement on whether measurements were taken from distinct samples or whether the same sample was measured repeatedly |
| ☐ | ☒ | The statistical test(s) used AND whether they are one- or two-sided<br>*Only common tests should be described solely by name; describe more complex techniques in the Methods section.* |
| ☒ | ☐ | A description of all covariates tested |
| ☒ | ☐ | A description of any assumptions or corrections, such as tests of normality and adjustment for multiple comparisons |
| ☐ | ☒ | A full description of the statistical parameters including central tendency (e.g. means) or other basic estimates (e.g. regression coefficient) AND variation (e.g. standard deviation) or associated estimates of uncertainty (e.g. confidence intervals) |
| ☐ | ☒ | For null hypothesis testing, the test statistic (e.g. *F*, *t*, *r*) with confidence intervals, effect sizes, degrees of freedom and *P* value noted<br>*Give P values as exact values whenever suitable.* |
| ☒ | ☐ | For Bayesian analysis, information on the choice of priors and Markov chain Monte Carlo settings |
| ☒ | ☐ | For hierarchical and complex designs, identification of the appropriate level for tests and full reporting of outcomes |
| ☐ | ☒ | Estimates of effect sizes (e.g. Cohen's *d*, Pearson's *r*), indicating how they were calculated |

*Our web collection on statistics for biologists contains articles on many of the points above.*

## Software and code

Policy information about availability of computer code

| | |
|---|---|
| Data collection | Image Studio v5.2 (LI-COR) |
| Data analysis | R version 4.0.2 (R Core Team, 2020), Bowtie 2 (Langmead and Salzberg, 2012), STAR v2.5.4 (Dobin et al., 2013), MACS2 v2.1.1 (Zhang et al., 2008), Sambamba (Tarasov et al., 2015), SAMtools v1.7 (Li et al., 2009), deepTools v3.1.1 (Ramirez et al., 2016), BEDtools v2.17.0 (Quinlan and Hall, 2010), DESeq2 (Love et al., 2014), Hi-C Pro 2.9 (Servant et al., 2015), GENOVA (van der Weide et al., 2021, github.com/deWitLab/GENOVA), HiCUP (Wingett et al., 2015), CHiCAGO (Cairns et al., 2016), CapSequm2 (Hughes et al., 2014), https://github.com/nFursova/Calibrated_ChIPseq_RNAseq |

For manuscripts utilizing custom algorithms or software that are central to the research but not yet described in published literature, software must be made available to editors and reviewers. We strongly encourage code deposition in a community repository (e.g. GitHub). See the Nature Portfolio guidelines for submitting code & software for further information.

## Data

Policy information about availability of data

All manuscripts must include a data availability statement. This statement should provide the following information, where applicable:
- Accession codes, unique identifiers, or web links for publicly available datasets
- A description of any restrictions on data availability
- For clinical datasets or third party data, please ensure that the statement adheres to our policy

The datasets generated in this study are available from GEO database under accession number GSE185930. Published data used in this study include mouse ESC TAD and loop coordinates (GSE96107), Polycomb domains (GSE119620), CDK8 peaks (GSE98756), and H3K27Ac peaks (GSE136424). For cnRNA-seq processing we

used mm10 (GenBank: BK000964.3, https://www.ncbi.nlm.nih.gov/nuccore/bk000964) and dm6 (GenBank: M21017.1, https://www.ncbi.nlm.nih.gov/nuccore/M21017.1) rDNA genomic datasets.

# Field-specific reporting

Please select the one below that is the best fit for your research. If you are not sure, read the appropriate sections before making your selection.

☒ Life sciences  ☐ Behavioural & social sciences  ☐ Ecological, evolutionary & environmental sciences

For a reference copy of the document with all sections, see nature.com/documents/nr-reporting-summary-flat.pdf

# Life sciences study design

All studies must disclose on these points even when the disclosure is negative.

| | |
|---|---|
| Sample size | Sample sizes were determined based on previous studies performed in the lab using similar techniques to enable reasonable statistical analysis. |
| Data exclusions | No data were excluded |
| Replication | Reported experimental findings were reproducible in multiple independent biological replicates. c-ChIPseq  and CapC experiments were performed in triplicates, cnRNA-seq - in quadruplicates, and Hi-C in duplicates. The numbers of biological replicates for each experiment are given in Methods section and/or in the figure legends. |
| Randomization | Randomization was not relevant for this study as it includes only molecular assays performed in cell lines of known genotype. |
| Blinding | Blinding was not relevant for this study as there were no prior assumptions about experimental outcomes. All data was collected and processed uniformly regardless of treatment groups. |

# Reporting for specific materials, systems and methods

We require information from authors about some types of materials, experimental systems and methods used in many studies. Here, indicate whether each material, system or method listed is relevant to your study. If you are not sure if a list item applies to your research, read the appropriate section before selecting a response.

## Materials & experimental systems

| n/a | Involved in the study |
|---|---|
| ☐ | ☒ Antibodies |
| ☐ | ☒ Eukaryotic cell lines |
| ☒ | ☐ Palaeontology and archaeology |
| ☒ | ☐ Animals and other organisms |
| ☒ | ☐ Human research participants |
| ☒ | ☐ Clinical data |
| ☒ | ☐ Dual use research of concern |

## Methods

| n/a | Involved in the study |
|---|---|
| ☐ | ☒ ChIP-seq |
| ☒ | ☐ Flow cytometry |
| ☒ | ☐ MRI-based neuroimaging |

# Antibodies

| | |
|---|---|
| Antibodies used | All anibodies used are listed in Supplementary Table |
| Validation | anti-CDK8 (ChIP-seq) - validated in Pelish et al 2015<br>anti-H3K27me3 - validated in Rose at al. 2016<br>anti-RING1B - manufacturer-validated against various cell lines by western blot, validated in the Klose lab in a conditional knock-out line by ChIP-seq and western blot (Fursova et al 2019)<br>anti-SUZ12 - manufacturer-validated against various cell lines by western blot, validated in the Klose lab in a conditional knock-out cell line (Dobrinic et al 2020)<br>anti-PCGF2 - manufacturer-validated, validated in the Klose lab in a conditional knock-out line by ChIP-seq and western blot (Fursova et al. 2019)<br>anti-CBX7 (ChIPseq) - manufacturer-validated, validated in the Klose lab in a conditional knock-out line by ChIP-seq and western blot (Fursova et al. 2019)<br>anti-CBX7 (Western blot) - manufacturer-validated by western blot, 7 citations: https://www.citeab.com/antibodies/221872-07-981-anti-cbx7-antibody?des=c3ab80f14feb824d<br>anti-T7-tag - validated here western blot with extracts from an untagged cell line by ChIPseq in a degron cell line (unpublished). Validated for ChIPseq in Brown et al 2017.<br>anti-TBP - manufacturer validated in various cell types by cellular fractionation, 235 citations: https://www.citeab.com/antibodies/753557-ab818-anti-tata-binding-protein-tbp-antibody-1tbp18?des=1ee5e4f398055d5b<br>anti-MED12 (54 citations https://www.citeab.com/antibodies/655615-a300-774a-rabbit-anti-med12-antibody-affinity-purifi? |

des=0e1bb45fbb1bfc17), anti-MED1 (92 citations https://www.citeab.com/antibodies/655647-a300-793a-rabbit-anti-med1-antibody-affinity-purifie?des=d802665adf8028ae), anti-MED14 (7 citations https://www.citeab.com/antibodies/654594-a301-044a-rabbit-anti-crsp2-drip150-antibody-affinit), anti-MED15 (6 citations https://www.citeab.com/antibodies/656694-a302-422a-rabbit-anti-med15-antibody-affinity-purifi?des=1f0dd4dd056c9dc6), anti-MED23 (13 citations https://www.citeab.com/antibodies/655005-a300-425a-rabbit-anti-crsp3-antibody-affinity-purifi?des=3dab3084bccb71ae) - manufacturer validated by IP and western blot
anti-MED17 - manufacturer validated by IF and Western blot using whole cell extract.
anti-MED13 - manufacturer validated by western blot and IP, validated here in a conditional knock-out cell line
anti-MED13L - manufacturer validated by western blot and IP, validated here in a conditional knock-out cell line
anti-CDK8 (Western blot) - validated by Western blot in the Klose lab in a knock-out cell line (unpublished)
anti-HDAC1 - manufacturer-validated by Western blot; 32 citations https://www.citeab.com/antibodies/762876-ab109411-anti-hdac1-antibody-epr460-2?des=ca96abb53494f759

## Eukaryotic cell lines

Policy information about cell lines

| Cell line source(s) | All mES cell lines used in this study were generated in the Klose lab:<br>CDK-MED cKO (Med13/13l fl/fl) mouse embryonic stem cell line (Dimitrova et al., 2018)<br>cPRC1 cKO (Pcgf4-/- Pcgf2 fl/fl) mouse embryonic stem cell line (Fursova et al., 2019)<br>TetR-PCGF2 TOT2N mouse embryonic stem cell line (with TetO integration) (Blackledge et al., 2014)<br>TetR-GFP TOT2N mouse embryonic stem cell line (with TetO integration) (Blackledge et al., 2014)<br>TetR-CDK8 TOT2N mouse embryonic stem cell line (with TetO integration) - this study<br>Med13/13l fl/fl Med14-T7 - mouse embryonic stem cell line generated in this study<br>Human HEK293T or drosophila SG4 cells (sourced from ATCC) were used as material for calibration but not as an experimental system. |
|---|---|
| Authentication | All cell lines generated in this study were validated by PCR, sequencing and Western blot. All cell lines generated for previous studies (Dimitrova et al, 2018; Blackledge et al, 2014; Fursova et al, 2019) were validated in their respective publication and confirmed in this study by Western blot. |
| Mycoplasma contamination | All cell lines were regularly tested for mycoplasma contamination and confirmed to be negative. |
| Commonly misidentified lines (See ICLAC register) | No commonly misidentified cell lines were used. |

## ChIP-seq

### Data deposition

☒ Confirm that both raw and final processed data have been deposited in a public database such as GEO.

☒ Confirm that you have deposited or provided access to graph files (e.g. BED files) for the called peaks.

| Data access links *May remain private before publication.* | https://www.ncbi.nlm.nih.gov/geo/query/acc.cgi?acc=GSE185930 |
|---|---|
| Files in database submission | 01-Med13fl-ESC-MED14-T7-UNT-rep1_R1.fastq.gz<br>01-Med13fl-ESC-MED14-T7-UNT-rep1_R2.fastq.gz<br>02-Med13fl-ESC-MED14-T7-TAM-rep1_R1.fastq.gz<br>02-Med13fl-ESC-MED14-T7-TAM-rep1_R2.fastq.gz<br>03-Med13fl-ESC-MED14-T7-UNT-rep2_R1.fastq.gz<br>03-Med13fl-ESC-MED14-T7-UNT-rep2_R2.fastq.gz<br>04-Med13fl-ESC-MED14-T7-TAM-rep2_R1.fastq.gz<br>04-Med13fl-ESC-MED14-T7-TAM-rep2_R2.fastq.gz<br>05-Med13fl-ESC-MED14-T7-UNT-rep3_R1.fastq.gz<br>05-Med13fl-ESC-MED14-T7-UNT-rep3_R2.fastq.gz<br>06-Med13fl-ESC-MED14-T7-TAM-rep3_R1.fastq.gz<br>06-Med13fl-ESC-MED14-T7-TAM-rep3_R2.fastq.gz<br>07-Med13fl-RA-MED14-T7-UNT-rep1_R1.fastq.gz<br>07-Med13fl-RA-MED14-T7-UNT-rep1_R2.fastq.gz<br>08-Med13fl-RA-MED14-T7-TAM-rep1_R1.fastq.gz<br>08-Med13fl-RA-MED14-T7-TAM-rep1_R2.fastq.gz<br>09-Med13fl-RA-MED14-T7-UNT-rep2_R1.fastq.gz<br>09-Med13fl-RA-MED14-T7-UNT-rep2_R2.fastq.gz<br>10-Med13fl-RA-MED14-T7-TAM-rep2_R1.fastq.gz<br>10-Med13fl-RA-MED14-T7-TAM-rep2_R2.fastq.gz<br>10-PCGF2-TAM-B1_R1.fastq.gz<br>10-PCGF2-TAM-B1_R2.fastq.gz<br>11-Med13fl-RA-MED14-T7-UNT-rep3_R1.fastq.gz<br>11-Med13fl-RA-MED14-T7-UNT-rep3_R2.fastq.gz<br>11-PCGF2-TAM-B2_R1.fastq.gz<br>11-PCGF2-TAM-B2_R2.fastq.gz<br>12-Med13fl-RA-MED14-T7-TAM-rep3_R1.fastq.gz<br>12-Med13fl-RA-MED14-T7-TAM-rep3_R2.fastq.gz |

```
12-PCGF2-TAM-B3_R1.fastq.gz
12-PCGF2-TAM-B3_R2.fastq.gz
13-Med13fl-ESC-Inp-UNT-rep1_R1.fastq.gz
13-Med13fl-ESC-Inp-UNT-rep1_R2.fastq.gz
14-Med13fl-ESC-Inp-TAM-rep1_R1.fastq.gz
14-Med13fl-ESC-Inp-TAM-rep1_R2.fastq.gz
15-Med13fl-ESC-Inp-UNT-rep2_R1.fastq.gz
15-Med13fl-ESC-Inp-UNT-rep2_R2.fastq.gz
16-Med13fl-ESC-Inp-TAM-rep2_R1.fastq.gz
16-Med13fl-ESC-Inp-TAM-rep2_R2.fastq.gz
17-Med13fl-ESC-Inp-UNT-rep3_R1.fastq.gz
17-Med13fl-ESC-Inp-UNT-rep3_R2.fastq.gz
18-Med13fl-ESC-Inp-TAM-rep3_R1.fastq.gz
18-Med13fl-ESC-Inp-TAM-rep3_R2.fastq.gz
19-Med13fl-RA-Inp-UNT-rep1_R1.fastq.gz
19-Med13fl-RA-Inp-UNT-rep1_R2.fastq.gz
1-RING1b-UNT-B1_R1.fastq.gz
1-RING1b-UNT-B1_R2.fastq.gz
20-Med13fl-RA-Inp-TAM-rep1_R1.fastq.gz
20-Med13fl-RA-Inp-TAM-rep1_R2.fastq.gz
21-Med13fl-RA-Inp-UNT-rep2_R1.fastq.gz
21-Med13fl-RA-Inp-UNT-rep2_R2.fastq.gz
22-Med13fl-RA-Inp-TAM-rep2_R1.fastq.gz
22-Med13fl-RA-Inp-TAM-rep2_R2.fastq.gz
23-Med13fl-RA-Inp-UNT-rep3_R1.fastq.gz
23-Med13fl-RA-Inp-UNT-rep3_R2.fastq.gz
24-Med13fl-RA-Inp-TAM-rep3_R1.fastq.gz
24-Med13fl-RA-Inp-TAM-rep3_R2.fastq.gz
25-INP-UNT-B1_R1.fastq.gz
25-INP-UNT-B1_R2.fastq.gz
26-INP-UNT-B2_R1.fastq.gz
26-INP-UNT-B2_R2.fastq.gz
27-INP-UNT-B3_R1.fastq.gz
27-INP-UNT-B3_R2.fastq.gz
28-INP-TAM-B1_R1.fastq.gz
28-INP-TAM-B1_R2.fastq.gz
29-INP-TAM-B2_R1.fastq.gz
29-INP-TAM-B2_R2.fastq.gz
2-RING1B-UNT-B2_R1.fastq.gz
2-RING1B-UNT-B2_R2.fastq.gz
31-INP-TAM-B3_R1.fastq.gz
31-INP-TAM-B3_R2.fastq.gz
35-Med13fl-RA-UNT-Cdk8-rep1_R1.fastq.gz
35-Med13fl-RA-UNT-Cdk8-rep1_R2.fastq.gz
36-Med13fl-RA-UNT-Cdk8-rep2_R1.fastq.gz
36-Med13fl-RA-UNT-Cdk8-rep2_R2.fastq.gz
3-RING1B-UNT-B3_R1.fastq.gz
3-RING1B-UNT-B3_R2.fastq.gz
42-Med13fl-RA-UNT-Cdk8-rep3_R1.fastq.gz
42-Med13fl-RA-UNT-Cdk8-rep3_R2.fastq.gz
4-RING1B-TAM-B1_R1.fastq.gz
4-RING1B-TAM-B1_R2.fastq.gz
5-RING1B-TAM-B2_R1.fastq.gz
5-RING1B-TAM-B2_R2.fastq.gz
6-RING1B-TAM-B3_R1.fastq.gz
6-RING1B-TAM-B3_R2.fastq.gz
7-PCGF2-UNT-B1_R1.fastq.gz
7-PCGF2-UNT-B1_R2.fastq.gz
8-PCGF2-UNT-B2_R1.fastq.gz
8-PCGF2-UNT-B2_R2.fastq.gz
9-PCGF2-UNT-B3_R1.fastq.gz
9-PCGF2-UNT-B3_R2.fastq.gz
lib18-Med13fl-UNT-Rep1-CDK8-ChIPseq_R1.fastq.gz
lib18-Med13fl-UNT-Rep1-CDK8-ChIPseq_R2.fastq.gz
lib20-Med13fl-TAM-Rep1-CDK8-ChIPseq_R1.fastq.gz
lib20-Med13fl-TAM-Rep1-CDK8-ChIPseq_R2.fastq.gz
lib21-Med13fl-UNT-Rep3-CDK8-ChIPseq_R1.fastq.gz
lib21-Med13fl-UNT-Rep3-CDK8-ChIPseq_R2.fastq.gz
lib22-Med13fl-TAM-Rep3-CDK8-ChIPseq_R1.fastq.gz
lib22-Med13fl-TAM-Rep3-CDK8-ChIPseq_R2.fastq.gz
lib23-Med13fl-UNT-RepB1-CDK8-ChIPseq_R1.fastq.gz
lib23-Med13fl-UNT-RepB1-CDK8-ChIPseq_R2.fastq.gz
lib24-Med13fl-TAM-RepB1-CDK8-ChIPseq_R1.fastq.gz
lib24-Med13fl-TAM-RepB1-CDK8-ChIPseq_R2.fastq.gz
MED13fl-K27me3-TAM-B1_R1.fastq.gz
MED13fl-K27me3-TAM-B1_R2.fastq.gz
```

```
MED13fl-K27me3-TAM-B2_R1.fastq.gz
MED13fl-K27me3-TAM-B2_R2.fastq.gz
MED13fl-K27me3-TAM-B3_R1.fastq.gz
MED13fl-K27me3-TAM-B3_R2.fastq.gz
MED13fl-K27me3-UNT-B1_R1.fastq.gz
MED13fl-K27me3-UNT-B1_R2.fastq.gz
MED13fl-K27me3-UNT-B2_R1.fastq.gz
MED13fl-K27me3-UNT-B2_R2.fastq.gz
MED13fl-K27me3-UNT-B3_R1.fastq.gz
MED13fl-K27me3-UNT-B3_R2.fastq.gz
MED13fl-NativeINP-TAM-B1_R1.fastq.gz
MED13fl-NativeINP-TAM-B1_R2.fastq.gz
MED13fl-NativeINP-TAM-B2_R1.fastq.gz
MED13fl-NativeINP-TAM-B2_R2.fastq.gz
MED13fl-NativeINP-TAM-B3_R1.fastq.gz
MED13fl-NativeINP-TAM-B3_R2.fastq.gz
MED13fl-NativeInp-UNT-B1_R1.fastq.gz
MED13fl-NativeInp-UNT-B1_R2.fastq.gz
MED13fl-NativeInp-UNT-B2_R1.fastq.gz
MED13fl-NativeInp-UNT-B2_R2.fastq.gz
MED13fl-NativeInp-UNT-B3_R1.fastq.gz
MED13fl-NativeInp-UNT-B3_R2.fastq.gz
PCGF2fl-TAM-Rep1-CDK8-ChIPseq_R1.fastq.gz
PCGF2fl-TAM-Rep1-CDK8-ChIPseq_R2.fastq.gz
PCGF2fl-TAM-Rep1-XInput_R1.fastq.gz
PCGF2fl-TAM-Rep1-XInput_R2.fastq.gz
PCGF2fl-TAM-Rep2-CDK8-ChIPseq_R1.fastq.gz
PCGF2fl-TAM-Rep2-CDK8-ChIPseq_R2.fastq.gz
PCGF2fl-TAM-Rep2-XInput_R1.fastq.gz
PCGF2fl-TAM-Rep2-XInput_R2.fastq.gz
PCGF2fl-TAM-Rep3-CDK8-ChIPseq_R1.fastq.gz
PCGF2fl-TAM-Rep3-CDK8-ChIPseq_R2.fastq.gz
PCGF2fl-TAM-Rep3-XInput_R1.fastq.gz
PCGF2fl-TAM-Rep3-XInput_R2.fastq.gz
PCGF2fl-UNT-Rep1-CDK8-ChIPseq_R1.fastq.gz
PCGF2fl-UNT-Rep1-CDK8-ChIPseq_R2.fastq.gz
PCGF2fl-UNT-Rep1-XInput_R1.fastq.gz
PCGF2fl-UNT-Rep1-XInput_R2.fastq.gz
PCGF2fl-UNT-Rep2-CDK8-ChIPseq_R1.fastq.gz
PCGF2fl-UNT-Rep2-CDK8-ChIPseq_R2.fastq.gz
PCGF2fl-UNT-Rep2-XInput_R1.fastq.gz
PCGF2fl-UNT-Rep2-XInput_R2.fastq.gz
PCGF2fl-UNT-Rep3-CDK8-ChIPseq_R1.fastq.gz
PCGF2fl-UNT-Rep3-CDK8-ChIPseq_R2.fastq.gz
PCGF2fl-UNT-Rep3-XInput_R1.fastq.gz
PCGF2fl-UNT-Rep3-XInput_R2.fastq.gz
MED1313Lfl_K27me3_TAM_mm10_spikeinnormalised_MERGED.MACS2.bw
MED1313Lfl_K27me3_UNT_mm10_spikeinnormalised_MERGED.MACS2.bw
MED1313Lfl_PCGF2_TAM_mm10_spikeinnormalised_MERGED_MACS2.bw
MED1313Lfl_PCGF2_UNT_mm10_spikeinnormalised_MERGED_MACS2.bw
MED1313Lfl_RING1B_TAM_mm10_spikeinnormalised_MERGED_MACS2.bw
MED1313Lfl_RING1B_UNT_mm10_spikeinnormalised_MERGED_MACS2.bw
Med13fl_ESC_TAM_MED14-T7_mm10_readcountnorm_MERGED_MACS2.bw.bw
Med13fl_ESC_UNT_MED14-T7_mm10_readcountnorm_MERGED_MACS2.bw.bw
Med13fl_RA_TAM_MED14-T7_mm10_readcountnorm_MERGED_MACS2.bw.bw
Med13fl_RA_UNT_MED14-T7_mm10_readcountnorm_MERGED_MACS2.bw.bw
mESC_MED13fl_TAM_CDK8_mm10.UniqMapped_downsampled_MERGED_MACS2.bw.bw
mESC_MED13fl_UNT_CDK8_mm10.UniqMapped_downsampled_MERGED_MACS2.bw.bw
mESC_Pcgf2fl_XChIP_TAM_CDK8_mm10.UniqMapped_downsampled_MERGED_MACS2.bw.bw
mESC_Pcgf2fl_XChIP_UNT_CDK8_mm10.UniqMapped_downsampled_MERGED_MACS2.bw.bw
Med13fl_RA_UNT_CDK8_mm10_spikeinnormalised.MERGED_downsampled.MACS2.bw
MED1313Lfl_K27me3_UNT_mm10_spikeinnormalised_MERGED.MACS2.bw
MED1313Lfl_K27me3_TAM_mm10_spikeinnormalised_MERGED.MACS2.bw
22_Med13fl_ESC_UNT_CBX7_ChIPseq_rep1_R1.fastq.gz
22_Med13fl_ESC_UNT_CBX7_ChIPseq_rep1_R2.fastq.gz
23_Med13fl_ESC_TAM_CBX7_ChIPseq_rep1_R1.fastq.gz
23_Med13fl_ESC_TAM_CBX7_ChIPseq_rep1_R2.fastq.gz
24_Med13fl_ESC_UNT_CBX7_ChIPseq_rep2_R1.fastq.gz
24_Med13fl_ESC_UNT_CBX7_ChIPseq_rep2_R2.fastq.gz
26_Med13fl_ESC_TAM_CBX7_ChIPseq_rep2_R1.fastq.gz
26_Med13fl_ESC_TAM_CBX7_ChIPseq_rep2_R2.fastq.gz
27_Med13fl_ESC_UNT_CBX7_ChIPseq_rep3_R1.fastq.gz
27_Med13fl_ESC_UNT_CBX7_ChIPseq_rep3_R2.fastq.gz
28_Med13fl_ESC_TAM_CBX7_ChIPseq_rep3_R1.fastq.gz
28_Med13fl_ESC_TAM_CBX7_ChIPseq_rep3_R2.fastq.gz
MED13fl_CBX7_UNT_mm10_spikeinnormalised_MERGED.MACS2.bw
```

MED13fl_CBX7_TAM_mm10_spikeinnormalised_MERGED.MACS2.bw

Genome browser session
(e.g. UCSC)

All merged bigWig files were uploaded to GEO: https://www.ncbi.nlm.nih.gov/geo/query/acc.cgi?acc=GSE185930

## Methodology

Replicates

All ChIP-seq experiments were performed in biological triplicates

Sequencing depth

All libraries were sequenced as 40bp paired-end reads. Number of reads is given in Supplementary Table.

Antibodies

Rabbit polyclonal anti-CDK8 Bethyl laboratories Cat# A302-500A lot 2
Rabbit monoclonal anti-RING1B  Cell Signaling Technology Cat# 5694 lot 3
Rabbit polyclonal anti-PCGF2 (Mel-18 H-115) Santa Cruz Cat# sc-10744 lot D0903
Rabbit polyclonal anti-H3K27me3 In house (Rose et al., 2016)
Rabbit monoclonal anti-T7-Tag (D9E1X) Cell Signaling Technology Cat# 13246 lot 1
Rabbit monoclonal anti-CBX7, abcam Cat#ab21873 lot GR3210651-1

Peak calling parameters

all peaks used in this study were published previously (Fursova et al, 2019; Dimitrova et al, 2018; Feldmann et al, 2020)

Data quality

Quality of ChIP-seq data was assessed by visual inspection of individual replicate bigWig files and comparison with other published data sets, as well as by metaplot, heatmap and correlation analysis using deepTools.

Software

Paired-end reads were aligned to the concatenated mouse (mm10) and spike-in (dm6 for native, hg19 for cross-linked cChIP-seq) genome sequences using Bowtie 2 ("–no-mixed" and "–no-discordant" options). Only uniquely mapped reads were kept for downstream analysis, after removal of PCR duplicates with Sambamba. Genome coverage tracks were generated using the pileup function from MACS2. Metaplot and heatmap analysis of ChIP-seq read density at regions of interest was performed with computeMatrix and plotProfile/plotHeatmap from deepTools.

