## [Peer Review File · Nature Structural & Molecular Biology]

Peer Review Information

Journal: Nature Structural and Molecular Biology

Manuscript Title: Distinct roles for CKM-Mediator in controlling Polycomb-dependent chromosomal interactions and priming genes for induction

Corresponding author name(s): Professor Robert Klose

Editorial Notes:

Reviewer Comments & Decisions:

Decision Letter, initial version:

28th Jan 2022

Dear Dr. Klose,

Thank you again for submitting your manuscript "Distinct roles for CDK-Mediator in controlling Polycomb-dependent chromosomal interactions and priming genes for induction". I apologize once again for the delayed reply. We now have comments (below) from the 3 reviewers who evaluated your paper. In light of those reports, we remain interested in your study and would like to see your response to the comments of the referees, in the form of a revised manuscript.

You will see that all 3 reviewers are positive about the potential interest of the findings, though each finds that additional experimentation or controls are necessary to clearly establish factor-factor interactions and gene occupancy in order to further support the proposed model. Reviewer #1 specifies assays and controls to clearly define interactions of CDK-Med and the Med core to establish that CDK-Med binds to poised genes independently and that the Med core is recruited upon induction. Reviewer #2 suggests assays to rule out the possibility that PRC1 recruits CDK-Med and that the factors are jointly required to establish interactions between Polycomb domains. Reviewer #3, Giacomo Cavalli, requests ChIP-seq analyses of additional PRC1 subunits to capture the full extent of PRC1 occupancy. He also queries whether CpG islands are additionally required for PRC1 recruitment and asks that this be tested using the tethering system. Lastly, he suggests controls for the PRC1 domain interaction assays and for verification that PRC1 is displaced from primed genes upon induction. Editorially, we agree that the above suggestions would strengthen the work, and ask that they be incorporated in a revised manuscript. While we appreciate Reviewer #1's recommendation of a rapid-depletion approach, we do not feel these assays are essential to support the central conclusions of the present work and would not ask that they be included.

Please be sure to address/respond to all concerns of the referees in full in a point-by-point response and highlight all changes in the revised manuscript text file. If you have comments that are intended for editors only, please include those in a separate cover letter.

We expect to see your revised manuscript within 6 weeks. If you cannot send it within this time, please contact us to discuss an extension; we would still consider your revision, provided that no similar work has been accepted for publication at NSMB or published elsewhere.

Reporting Summary:

When submitting the revised version of your manuscript, please pay close attention to our [href="https://www.nature.com/nature-research/editorial-policies/image-integrity">Digital Image Integrity Guidelines.](https://www.nature.com/nature-research/editorial-policies/image-integrity) and to the following points below:

- that unprocessed scans are clearly labelled and match the gels and western blots presented in figures.
- that control panels for gels and western blots are appropriately described as loading on sample

processing controls

-- all images in the paper are checked for duplication of panels and for splicing of gel lanes.

Please note that all key data shown in the main figures as cropped gels or blots should be presented in uncropped form, with molecular weight markers. These data can be aggregated into a single supplementary figure item. While these data can be displayed in a relatively informal style, they must refer back to the relevant figures. These data should be submitted with the final revision, as source data, prior to acceptance, but you may want to start putting it together at this point.

Data availability: this journal strongly supports public availability of data. All data used in accepted papers should be available via a public data repository, or alternatively, as Supplementary Information. If data can only be shared on request, please explain why in your Data Availability Statement, and also in the correspondence with your editor. Please note that for some data types, deposition in a public repository is mandatory - more information on our data deposition policies and available repositories can be found below:

<https://www.nature.com/nature-research/editorial-policies/reporting-standards#availability-of-data>

Deposition of deep sequencing and microarray data is mandatory, and the datasets must be released prior to or upon publication. To avoid delays in publication, dataset accession numbers must be supplied with the final accepted manuscript and appropriate release dates must be indicated at the galley proof stage.

[Redacted]

With kind regards,

Beth

Beth Moorefield, Ph.D.
Senior Editor
Nature Structural & Molecular Biology

Referee expertise:

Referee #1: Mediator function

Referee #2: Genome organization/differentiation

Referee #3: Self-identifies

Reviewers' Comments:

Reviewer #1:

Remarks to the Author:

The role of the four-subunit kinase module (CKM) of the Mediator has long been enigmatic. The earliest biochemical studies on both the yeast and the metazoan versions of the complex strongly hinted at a negative role for this module. While these studies suggested inhibitory mechanisms that included outright occlusion of Mediator-RNA polymerase II (Pol II) interactions and phosphorylation of a key general factor (TFIIH), cell-based studies also suggested roles of this module in long-term ("epigenetic") silencing. At the same time, other cell-based studies hinted at roles that were more consistent with an overall positive role for the module in gene activation. In this manuscript the authors have revisited what they have dubbed a "yin-and-yang" role for the CKM module (which they term "CDK"; see below). The authors come to the problem from their studies on the interactions prevailing at inactive gene regulatory sites that are bound by repressive Polycomb (PRC) complexes in embryonic stem cells (ESCs) and from prior indications that CKM-Mediator might prime genes for induction during differentiation. Through inducible knock out (KO) of its MED13/MED13L subunits, which partially disrupts the subcomplex and appears to prevent its interaction with the core Mediator complex, they find that the CKM module contributes very little to the 3D genome organization in ESCs. However, the module somehow regulates the binding of PRC, and thus interactions between Polycomb domains; removal of PRC (PCGF2/4 KO) specifically disrupts these interactions. This allowed the authors to design a series of "separation-of-function" experiments that suggest to them that CKM-Mediator can prime gene induction independently of Polycomb domain interactions. Additional ChIP experiments led them to propose a model in which the CKM module can bind to inactive developmental gene promoters independently of the core Mediator, the latter only being recruited (via the CKM module) once the genes have been activated (here through stimulation of ESCs by retinoic acid).

The manuscript advances a number of provocative ideas, but unfortunately, the experimental support for these remains rather weak. As itemized in greater detail below, the paper has three major issues that in this reviewer's opinion need to be addressed for the paper to be the substantial contribution that it can be. First, some of the observations, especially those pertaining to how CKM-Mediator might facilitate PRC binding and resultant domain interactions, do not go beyond simple phenomenological

descriptions. In that sense the study is somewhat premature. Second, general methodological problems (long-duration inducible KO; incomplete ChIP analyses and lack of rigorous ChIP controls) undercut the authors' key conclusions and proposals. Finally, and perhaps related to the first two points, at various points in the text there is confusion as to whether the authors are describing just the CKM or the CKM-Mediator. This makes for great difficulty in following what exactly the authors may have in mind.

Specific points:

1. That CKM-Mediator facilitates binding of PRC to engender Polycomb domain interactions is an intriguing finding. Yet, the issue remains under-developed as no mechanistic insights, or even clues into the mechanism, are provided. Indeed, it remains unclear whether the contribution of CKM-Mediator in this phenomenon is direct or indirect. Whereas the authors understandably relied on widely used inducible KO methods to deplete MED13/13L, this particular study would have greatly benefitted from newer technologies that are based on acute degradation of targeted proteins (e.g., via dTAG) and thus are able to provide readouts on its direct (immediate) effects (or not!) on gene transcription.
2. In revising the manuscript, the authors must pay careful attention to defining the kinase module consisting of CDK8/19, CNCC, MED12/12L, and MED13/MED13L on the one hand and its complex with core Mediator on the other hand. Currently, there appears to be significant confusion between these two distinct entities. The authors do introduce "CDK-MED" (Introduction, line 62). But its composition is not clarified and the particular nomenclature the authors opted for is not very helpful. While this will ultimately be an editorial decision, this reviewer prefers to use more descriptive terms like "CKM" for the kinase module and "CKM-Mediator" (with Mediator fully spelled out) for the complex containing the module.
3. The nomenclature issue aside, there is a serious problem in the manuscript with regard to accurately gauging the dynamics between the CKM module and the core Mediator. This is attributable in large part to absence of certain controls and how the authors chose to score for the core Mediator. Please note the following suggestions:
 - (i) To substantiate the schematic depiction of the residual complex in Fig. 1A, please re-probe CDK8 IPs in Sup Fig. 1A from MED13/13L KO with a CCNC antibody.
 - (ii) Also in Sup Fig. 1A, please minimally include some representative subunits from the head and middle modules. Given their tail/tail-proximal locations, and potential lability, MED15 and MED1 may not be the ideal subunits to probe here. A MED14 blot should definitely be included.
 - (iii) Perhaps most crucially, the ChIP assays in Fig. 5/Sup Fig. 5 need significant new controls to make the data and the conclusions credible. Not having recourse to a ChIP-grade MED14 antibody, the authors added 3XT7 and 2XStrepII-FKBP epitope tags to the endogenous Med14 gene to score for the backbone MED14 subunit. This raises the question as to whether a core Mediator containing this tag at the N-terminus of MED14, close to where the kinase module might interact, is in fact capable of being scored in the context of a CKM-Mediator complex. Absent controls to eliminate this concern, the authors' conclusion that in uninduced ESCs, poised genes are occupied by the 4-subunit kinase module, which upon induction is replaced by the core Mediator is simply not tenable. As also discussed in Point 4 below, this reviewer remains unconvinced that the authors have ruled out a scenario in which the CKM-Mediator is first recruited and then "evicts" the kinase module upon gene activation. The authors might argue that the data in Sup Fig. 5D would serve as a control. However, at present, a straightforward explanation for the authors' ChIP data is that in the more dynamic scenario of H3K27ac+ genes CKM might be coming on and off the core Mediator, making it feasible to detect both CDK8 and MED14 (T7). By contrast, on the relatively more static situation of poised PcG genes the tag

may not be properly scored. Therefore, control experiments must include: (a) as a proxy for the ChIP experiments, a demonstration that the T7 antibody (D9E1X) does in fact quantitatively IP both a 3XT7-2XStrepII-FKBP-tagged core Mediator and the same in association with the kinase module; and (b) ChIP data (Fig. 5A) for other core subunits (again, ideally representative of all major modules), especially if properly scoring MED14 turns out to be problematic.

4. Further with regard to the authors' contention that the kinase module might be recruited to poised genes independently of the core Mediator, they cite Pavri/Reinberg 2005 (Ref 58) as providing a precedent. This appears to be a misreading of the findings in that paper. Fig. 6F therein very clearly shows that MED6 (core Mediator) constitutively (before and after induction) occupies the target gene whereas CDK8 (kinase module) is present only in the inactive state. This result is more compatible with a more plausible model in which the kinase module is evicted upon induction. Please amend the text to reflect this, even as it is anticipated that once the authors do the suggested control experiments (Point 3), they might yet come to the actual conclusion of Ref 58!

5. More on incorrect references: as written, the statement in lines 58-61 seems to suggest that references 41 through 46 all contributed to establishing that the kinase module is associated with genes in ESCs. Since the cited papers focus on many distinct aspects of CKM function, please rephrase to appropriately credit the contributions of individual papers.

Reviewer #2:

Remarks to the Author:

This is an elegant study that provides novel and important insights into the roles of Polycomb group and Mediator complexes in developmental gene expression control. Both these complexes had been implicated in establishing long-range chromosomal interactions in pluripotent stem cells, and in priming developmental genes for activation during differentiation. This study makes key contributions towards disentangling these proposed functions. The authors first show that long-range chromosomal interactions between PcG bound regions are dramatically weakened in the absence of the CDK-MED complex. Loss of CDK-MED results in reduced binding of the canonical Polycomb group complex 1 (PRC1) while only weakly affecting H3K27me occupancy at PcG sites. Recruiting PRC1, but not CDK-MED, to an ectopic site in the genome establishes long-range interactions with endogenous Polycomb-bound regions. Interestingly, the PcG-mediated interactions do not appear to play a major role (as had been speculated previously) in priming the gene loci involved for activation during development. Instead, CDK-MED binding enables core Mediator complex binding to these loci to facilitate gene activation.

This is an original study with high quality data that is adequately described and discussed in the context of the current literature. The approach is very logical and systematic, and the conclusions are robust, with one exception (see major comment below), where I think an additional control experiment is required to support the authors' conclusions. The language is clear and precise, but there appear to be some mistakes in the figures that will need to be corrected (see minor comments below). Overall, a well executed study with important novel insights into the relationship between 3D genome organisation and function. Before I could recommend publication, I would like to see the following concerns/comments addressed:

1. Major comments:

The system using recruitment to an ectopic TetO operator (Figure 3) is very elegant and builds on the

previous use of this system in the Klose lab to dissect Polycomb group protein recruitment hierarchies. However, can the authors exclude the possibility that PCGF2-TetR recruitment results in the concomitant recruitment of CDK-MED to the ectopic site? And thus, that an interplay between PRC1 and CDK-MED is required to establish and/or maintain chromosomal interactions between Polycomb domains? In light of other results presented in this study, this is unlikely but should nonetheless be addressed directly. ChIP-qPCR for CDK-MED in the ESCs with ectopic PCGF2-TetR recruitment would answer this question, and given the expertise in the lab, should be a relatively straightforward experiment.

Lines 105-106: "Therefore, we discover the CDK-MED is essential for interactions between Polycomb domains."

Is this true across all PcG interactions? Are there some that remain unchanged in the absence of CDK-MED? Are some interactions between PcG-bound regions more dramatically affected than others? If yes, does this correlate with CDK-MED binding?

2. Minor comments:

Figure 1F: clearly shows reduction of PcG dependent interactions over relatively short-range distances (50 kb) – how about longer-range interactions?

Aggregate plot with different distance range would be informative (up to 500 kb or 1 Mb). The examples in Figures 1E and 1G suggest that long-range interactions should be equally affected, but it would be nice to see an aggregate plot that directly assesses the effect of CDK-MED depletion on longer-range interactions.

Figure 1G: please add genome coordinates of the window shown in the legend. Is this locus on mouse chromosome 12? If yes, should this be Pax9 instead of Pax2? See also Figure 2B.

Figure 3D: see above comments on Figure 1G – please add genomic coordinates here and ensure that the genes are correctly labelled (Pax2 or Pax9)?

Figure 4E: is the 500kb scale bar correct? According to the genome coordinates, a ~2.7Mb window is shown here. To my eye, it just doesn't look like the 500 kb bar would fit over five times into this screenshot. I suspect either the 500kb bar or the genomic coordinates are wrong.

Line 277: should be 'Mediator binding' instead of 'Mediator biding'

Line 800: "...showing deletion of the MED13 and MED13L proteins." Should this be depletion rather than deletion?

Reviewer #3:

Remarks to the Author:

Review of NSMB-A45650

In this work, the authors analyzed the role of a specific form of Mediator (CDK-MED) in Polycomb function and in priming gene expression in ES cells prior to or upon retinoic acid mediated differentiation. They report that upon loss of CDK-MED, cPRC1-mediated long-range chromatin

interactions are lost, whereas little other changes in 3D genome folding are detected. Loss of cPRC1-dependent interactions depends on reduction of cPRC1 occupancy at their targets. They then set up separation of function experiments, showing that tethering cPRC1 to a naïve genomic site induces 3D interactions with other cPRC1 sites, whereas CDK-MED tethering does not induce the same. Furthermore, CDK-MED mediates priming of gene induction during retinoic acid induced differentiation but this does not require cPRC1 interactions. Instead, it correlates with recruitment of the core mediator (tracked by mapping its MED14 subunit).

The work is well carried out and the data are very clean. It would ultimately bring important facts to the table that would be of interest for a broad audience. However, I think that the authors should address the following points:

- 1) The authors study binding of cPRC1 upon loss of CDK-MED. To do so, they do cChIP-seq of the RING1B subunit (which however marks all PRC1 complexes including variants) and the cPRC1 specific subunit PCGF2. However, PCGF4/Bmi1 can also be present in cPRC1, therefore looking at PCGF2 might not reveal all of cPRC1. I ask the authors to do cChIP-seq CBX7, which makes for the overwhelming majority of PRC1 in ESCs and would be an important complementation of the current data.
- 2) In figure 3 the authors tether either cPRC1 or CDK-MED to a naïve site and show that the former induces long-range interactions with other PRC1 sites, whereas the latter does not recruit PRC1, nor does it interact with PRC1-bound sites. Clear result, but why would CDK-MED be necessary but not sufficient for cPRC1 recruitment? Does recruitment need an additional CpG island? It might be interesting to insert such a CpG island flanking the TetO site in order to test for sufficiency. Furthermore, do the endogenous CDK-MED sites unbound by cPRC1 correspond to non-CpG island promoters?
- 3) In figure 1H, 3E and S3I the authors show interactions between PRC1 bound promoters with other Polycomb domains and use as controls interactions between non Polycomb promoters and active sites. First, are these pairs of sites distance-matched? If not, they should be. Second, it would be interesting to study the interactions of inactive non-Polycomb promoters with distance-matched regions devoid of active sites in order to provide a control of the behaviour of inactive chromatin that is not PRC1 dependent.
- 4) In Fig 4 the authors show that CDK-MED primes genes for activation during retinoic acid (RA) induced cell differentiation. They show that PRC1 depletion has no effect on these genes. However, what they do not show is whether cPRC1 is displaced from CDK-MED primed genes after addition of RA in wt cells and whether in the absence of CDK-MED cPRC1 might stay bound on these genes.
- 5) concerning the experiment in Figure 5 with MED14 and related to my last point, it would be very interesting to deplete MED14 during RA-induced differentiation to see whether cPRC1 binding to CDK-MED primed genes stays higher than in the presence of wt MED14.

Giacomo Cavalli

Author Rebuttal to Initial comments

Decision Letter, first revision:

Our ref: NSMB-A45650A

12th Apr 2022

Dear Rob,

Thank you for submitting your revised manuscript "Distinct roles for CKM-Mediator in controlling Polycomb-dependent chromosomal interactions and priming genes for induction" (NSMB-A45650A). It has now been seen by the original referees and their comments are below. The reviewers find that the paper has improved in revision, and that their prior concerns have been addressed. We'll be happy in principle to publish it in Nature Structural & Molecular Biology, pending minor revisions to comply with our editorial and formatting guidelines.

To facilitate our work at this stage, we would appreciate if you could send us the main text as a Word file. Please make sure to copy the NSMB account (cc'ed above).

With kind regards,

Beth

Beth Moorefield, Ph.D.
Senior Editor
Nature Structural & Molecular Biology

Reviewer #1 (Remarks to the Author):

In this revised version of their manuscript aiming to implicate the kinase module (CKM) of the Mediator in priming genes for induction, the authors have made a number of important changes that have noticeably improved it. This reviewer was previously mainly concerned that the authors' central thesis relating to a role for the CKM module of the Mediator complex remained unclear, both because of inconsistent terminology used in the text and incomplete documentation of the phenomenology being reported. The text is now much clearer as to what the authors are trying to claim, and the newly added data/controls help to dispel some of the original concerns.

Happily, the authors did the suggested important control IP to validate the T7 tag that they put on MED14 (Fig 5SD). Their claims would have been further strengthened had they opted to include additional antibodies against some core Mediator subunits in the ChIP experiments, as suggested. This would have given a comprehensive readout of what is happening and removed any lingering concerns

that reliance on a solitary epitope may not be an authentic indicator of core Mediator complex dynamics in the context of chromatin. In their rebuttal, the authors justify their decision to not attempt these additional ChIP experiments partly on the basis of results from the El Khattabi paper, which nonetheless lists at least two antibodies (MED23 and MED26) that worked well. The El Khattabi finding on MED1 notwithstanding, please also note also that several labs (including R. Young) have routinely shown nice ChIP data for MED1.

Finally, in relation to an earlier raised point, it is totally understandably that the authors were unable to do degron-type experiments for this paper. However, I think that somewhere an explicit acknowledgement should be made of the of the caveat that there is a chance that the authors may be looking at indirect effects of knocking out MED13.

Reviewer #2 (Remarks to the Author):

The authors have fully addressed the points I raised, and I am now happy to recommend this study for publication.

Reviewer #3 (Remarks to the Author):

The authors have addressed my comments and, in my opinion, those of other reviewers as well. I approve publication of the manuscript in this form.
Giacomo Cavalli

Decision Letter, final checks:

Our ref: NSMB-A45650A

13th May 2022

Dear Dr. Klose,

Thank you for your patience as we've prepared the guidelines for final submission of your Nature Structural & Molecular Biology manuscript, "Distinct roles for CKM-Mediator in controlling Polycomb-dependent chromosomal interactions and priming genes for induction" (NSMB-A45650A). Please carefully follow the step-by-step instructions provided in the attached file, and add a response in each row of the table to indicate the changes that you have made. Please also check and comment on any additional marked-up edits we have proposed within the text. Ensuring that each point is addressed will help to ensure that your revised manuscript can be swiftly handed over to our production team.

In recognition of the time and expertise our reviewers provide to Nature Structural & Molecular Biology's editorial process, we would like to formally acknowledge their contribution to the external peer review of your manuscript entitled "Distinct roles for CKM-Mediator in controlling Polycomb-dependent chromosomal interactions and priming genes for induction". For those reviewers who give their assent, we will be publishing their names alongside the published article.

Nature Structural & Molecular Biology offers a Transparent Peer Review option for new original research manuscripts submitted after December 1st, 2019. As part of this initiative, we encourage our authors to support increased transparency into the peer review process by agreeing to have the reviewer comments, author rebuttal letters, and editorial decision letters published as a Supplementary item. When you submit your final files please clearly state in your cover letter whether or not you would like to participate in this initiative. Please note that failure to state your preference will result in delays in accepting your manuscript for publication.

Cover suggestions

As you prepare your final files we encourage you to consider whether you have any images or illustrations that may be appropriate for use on the cover of Nature Structural & Molecular Biology.

Nature Structural & Molecular Biology has now transitioned to a unified Rights Collection system which will allow our Author Services team to quickly and easily collect the rights and permissions required to publish your work. Approximately 10 days after your paper is formally accepted, you will receive an email in providing you with a link to complete the grant of rights. If your paper is eligible for Open Access, our Author Services team will also be in touch regarding any additional information that may be required to arrange payment for your article.

Please note that *Nature Structural & Molecular Biology* is a Transformative Journal (TJ). Authors may publish their research with us through the traditional subscription access route or make their paper immediately open access through payment of an article-processing charge (APC). Authors

will not be required to make a final decision about access to their article until it has been accepted. [Find out more about Transformative Journals](https://www.springernature.com/gp/open-research/transformative-journals)

Authors may need to take specific actions to achieve [compliance with funder and institutional open access mandates](https://www.springernature.com/gp/open-research/funding/policy-compliance-faqs). If your research is supported by a funder that requires immediate open access (e.g. according to [Plan S principles](https://www.springernature.com/gp/open-research/plan-s-compliance)) then you should select the gold OA route, and we will direct you to the compliant route where possible. For authors selecting the subscription publication route, the journal's standard licensing terms will need to be accepted, including [self-archiving policies](https://www.springernature.com/gp/open-research/policies/journal-policies). Those licensing terms will supersede any other terms that the author or any third party may assert apply to any version of the manuscript.

Please use the following link for uploading these materials:
[Redacted]

Best regards,

Sophia Frank
Editorial Assistant
Nature Structural & Molecular Biology
nsmb@us.nature.com

On behalf of

Carolina Perdigoto, PhD
Chief Editor
Nature Structural & Molecular Biology

Reviewer #1:

Remarks to the Author:

In this revised version of their manuscript aiming to implicate the kinase module (CKM) of the Mediator in priming genes for induction, the authors have made a number of important changes that have noticeably improved it. This reviewer was previously mainly concerned that the authors' central thesis relating to a role for the CKM module of the Mediator complex remained unclear, both because of inconsistent terminology used in the text and incomplete documentation of the phenomenology being reported. The text is now much clearer as to what the authors are trying to claim, and the newly added data/controls help to dispel some of the original concerns.

Happily, the authors did the suggested important control IP to validate the T7 tag that they put on MED14 (Fig 5SD). Their claims would have been further strengthened had they opted to include additional antibodies against some core Mediator subunits in the ChIP experiments, as suggested. This would have given a comprehensive readout of what is happening and removed any lingering concerns that reliance on a solitary epitope may not be an authentic indicator of core Mediator complex dynamics in the context of chromatin. In their rebuttal, the authors justify their decision to not attempt these additional ChIP experiments partly on the basis of results from the El Khattabi paper, which nonetheless lists at least two antibodies (MED23 and MED26) that worked well. The El Khattabi finding on MED1 notwithstanding, please also note also that several labs (including R. Young) have routinely shown nice ChIP data for MED1.

Finally, in relation to an earlier raised point, it is totally understandably that the authors were unable to do degron-type experiments for this paper. However, I think that somewhere an explicit acknowledgement should be made of the of the caveat that there is a chance that the authors may be looking at indirect effects of knocking out MED13.

Reviewer #2:**Remarks to the Author:**

The authors have fully addressed the points I raised, and I am now happy to recommend this study for publication.

Reviewer #3:**Remarks to the Author:**

The authors have addressed my comments and, in my opinion, those of other reviewers as well. I approve publication of the manuscript in this form.

Giacomo Cavalli

Final Decision Letter:

Dear Dr. Klose,

We are now happy to accept your revised paper "Distinct roles for CKM-Mediator in controlling

Polycomb-dependent chromosomal interactions and priming genes for induction" for publication as a Article in Nature Structural & Molecular Biology.

As soon as your article is published, you can generate your shareable link by entering the DOI of your article here: http://authors.springernature.com/share.

Corresponding authors will also receive an automated email with the shareable link

Note the policy of the journal on data deposition:

<http://www.nature.com/authors/policies/availability.html>.

Your paper will be published online soon after we receive proof corrections and will appear in print in the next available issue. You can find out your date of online publication by contacting the production team shortly after sending your proof corrections. Content is published online weekly on Mondays and Thursdays, and the embargo is set at 16:00 London time (GMT)/11:00 am US Eastern time (EST) on the day of publication. Now is the time to inform your Public Relations or Press Office about your paper, as they might be interested in promoting its publication. This will allow them time to prepare an accurate and satisfactory press release. Include your manuscript tracking number (NSMB-A45650B) and our journal name, which they will need when they contact our press office.

About one week before your paper is published online, we shall be distributing a press release to news organizations worldwide, which may very well include details of your work. We are happy for your institution or funding agency to prepare its own press release, but it must mention the embargo date

and Nature Structural & Molecular Biology. If you or your Press Office have any enquiries in the meantime, please contact press@nature.com.

Please note that *Nature Structural & Molecular Biology* is a Transformative Journal (TJ). Authors may publish their research with us through the traditional subscription access route or make their paper immediately open access through payment of an article-processing charge (APC). Authors will not be required to make a final decision about access to their article until it has been accepted. [Find out more about Transformative Journals](https://www.springernature.com/gp/open-research/transformative-journals)

Authors may need to take specific actions to achieve [compliance with funder and institutional open access mandates](https://www.springernature.com/gp/open-research/funding/policy-compliance-faqs). If your research is supported by a funder that requires immediate open access (e.g. according to [Plan S principles](https://www.springernature.com/gp/open-research/plan-s-compliance)) then you should select the gold OA route, and we will direct you to the compliant route where possible. For authors selecting the subscription publication route, the journal's standard licensing terms will need to be accepted, including [self-archiving policies](https://www.springernature.com/gp/open-research/policies/journal-policies). Those licensing terms will supersede any other terms that the author or any third party may assert apply to any version of the manuscript.

Sincerely,

Carolina Perdigoto, PhD
Chief Editor
Nature Structural & Molecular Biology
orcid.org/0000-0002-5783-7106
